# Type I interferons affect the metabolic fitness of CD8+ T cells from patients with systemic lupus erythematosus

Norzawani Buang[1], Lunnathaya Tapeng[1], Victor Gray[1], Alessandro Sardini[2], Chad Whilding[2], Liz Lightstone[1,3], Thomas D. Cairns[3], Matthew C. Pickering[1,3], Jacques Behmoaras[1], Guang Sheng Ling [1,4,5 ✉] & Marina Botto [1,3,5 ✉]

The majority of patients with systemic lupus erythematosus (SLE) have high expression of type I IFN-stimulated genes. Mitochondrial abnormalities have also been reported, but the contribution of type I IFN exposure to these changes is unknown. Here, we show down-regulation of mitochondria-derived genes and mitochondria-associated metabolic pathways in IFN-High patients from transcriptomic analysis of CD4+ and CD8+ T cells. CD8+ T cells from these patients have enlarged mitochondria and lower spare respiratory capacity associated with increased cell death upon rechallenge with TCR stimulation. These mitochondrial abnormalities can be phenocopied by exposing CD8+ T cells from healthy volunteers to type I IFN and TCR stimulation. Mechanistically these 'SLE-like' conditions increase CD8+ T cell NAD+ consumption resulting in impaired mitochondrial respiration and reduced cell viability, both of which can be rectified by NAD+ supplementation. Our data suggest that type I IFN exposure contributes to SLE pathogenesis by promoting CD8+ T cell death via metabolic rewiring.

[1] Department of Immunology and Inflammation, Centre for Inflammatory Disease, Imperial College London, London, UK. [2] MRC London Institute of Medical Sciences, Imperial College London, London, UK. [3] Imperial Lupus Centre, Imperial College Healthcare NHS Trust, London, UK. [4] Present address: School of Biomedical Sciences, LKS Faculty of Medicine, The University of Hong Kong, Hong Kong, China. [5] These authors contributed equally: Guang Sheng Ling, Marina Botto. ✉email: gsling@hku.hk; m.botto@imperial.ac.uk

Systemic lupus erythematosus (SLE) is a relapsing-remitting autoimmune disease for which we lack biological parameters to monitor and predict disease flares. A large body of research has demonstrated that the majority of SLE patients have an increased expression of type I IFN-stimulated genes (ISGs), known as "IFN signature"[1]. Type I IFNs, which consist of IFN-α, IFN-β, IFN-ε, IFN-κ, and IFN-ω, are potent pleiotropic cytokines with multiple and diverse immune functions. Increased expression of ISGs has been reported to correlate with SLE disease activity in cross-sectional studies[2]. However, studies focussing on ISG expression in cells from active and inactive SLE patients and longitudinal studies have reported contradictory results with a robust correlation between ISG levels and flare rates in paediatric[3], but not in adult patients[4]. Several gene products encoded by ISGs are linked to antiviral defence mechanisms and cellular function regulation[5,6]. Nonetheless, the biological role(s) of these genes in SLE remains poorly understood. The triggers of the IFN signature in SLE are also not well-defined. There are at least three possible causes. Firstly, immune complexes, as well as the DNA and RNA-containing materials, can act as endogenous IFN inducers by activating plasmacytoid dendritic cells (pDCs) via the TLR7/TLR8 or TLR9 signaling pathway[7]. Secondly, genetic variants within the type I IFN signaling pathway can enhance IFN-α production[8]. Finally, defects in the regulation of pDC activation can result in continuous activation of these cells by endogenous nucleic acids[6].

Some studies suggest that in addition to nutrient, oxygen, and growth factor availability, cytokines can also affect the mitochondrial metabolism and dynamics[9]. Type I IFNs have been shown to affect mitochondrial functions in many cell types[10]. Type I IFNs activate mitochondrial apoptotic pathways in hematopoietic cells by increasing mitochondrial reactive oxygen species (ROS) accumulation[11]. Furthermore, IFNα-treated monocytes display defective mitophagy and cytoplasmic accumulation of mitochondrial DNA (mtDNA), leading to an increased inflammatory phenotype[12]. Paradoxically, some studies have shown that type I IFNs downregulate mtDNA-encoded gene expression, reduce electron transport chain (ETC) activity and exert an antiproliferative effect in human B cell lines[13]. Furthermore, type I IFNs promote mitochondrial functions in pDCs through upregulation of fatty acid oxidation (FAO) and oxidative phosphorylation (OXPHOS)[14]. The same study also demonstrates the type I IFNs promote OXPHOS in other immune cells, including murine memory CD8+ T cells, but not effector CD8+ T cells. Collectively, these studies suggest that the downstream effects of type I IFNs on mitochondrial metabolism vary and are cell-specific.

Several studies have described mitochondrial and metabolic alterations in cells from SLE patients. Common variants in mtDNA-encoded genes are reported to be associated with SLE[15] and SLE leukocytes have increased mtDNA damage and decreased mtDNA-encoded gene expression[16]. Metabolic changes have been primarily reported in CD4+ T cells and include mitochondrial hyperpolarization, ATP reduction, increased ROS production, intracellular depletion of the anti-oxidant glutathione[17], and enhanced glycolysis[18–20]. These metabolic changes lead to abnormal T cell activation and impaired cell-death pathways, which can contribute to the immune dysregulation and autoantibody production[21]. In keeping with these, we found that complement C1q, a key susceptibility gene in SLE, limits the autoimmune response by acting as a metabolic regulator of CD8+ T cells[22]. Despite the accumulating evidence of a role of type I IFNs in immune cell metabolism, most of the data so far are derived from in vitro experiments with short-term exposure or murine models. At present, the effects of chronic type I IFN signaling on human T cell metabolism remain unclear.

In this study, using high-throughput RNA sequencing in T cells from SLE patients, we show a link between type I IFN signature and mitochondrial changes in CD8+ T cells. We demonstrate that the metabolic abnormalities seen in the CD8+ T cells are the result of the combined prolonged type I IFN exposure and TCR stimulation. Together these two stimuli increase the consumption of nicotinamide adenine dinucleotide (NAD+), alter the morphology and function of the mitochondria in CD8+ T cells, making them metabolically unfit and more prone to die, especially under stress conditions or in response to an antigen rechallenge.

## Results

**Downregulation of OXPHOS genes in patients with high expression of ISGs**. To investigate if T cell transcriptomic signatures could help determining disease activity in SLE patients, we sequenced mRNA in CD4+ and CD8+ T cells isolated from active and inactive female patients (see "Methods" section and Supplementary Table 1). The T cell signatures were compared to a cohort of age-matched and sex-matched healthy controls (HC, $n = 11$). Differential gene expression analysis revealed minimal or few differentially expressed genes (DEGs) between active and inactive SLE patients, whilst the comparison between all SLE patients and HC showed more DEGs in CD8+ than in CD4+ T cells (358 and 237 genes respectively; Supplementary Table 2). As expected, pathway analyses uncovered highly enriched IFN signaling pathways in both SLE CD4+ and CD8+ T cell transcriptomes. We also found pathways that had been previously described to be over-expressed in SLE transcriptome[23,24], such as JAK-STAT signaling, cell cycle, DNA repair, and apoptosis (Supplementary Fig. 1).

We next performed hierarchical clustering of all CD4+ and CD8+ T cell DEGs between healthy controls and SLE patients, irrespective of the patient clinical features. This analysis segregated SLE patients into two groups: SLE-1 and SLE-2 (Fig. 1a, b). In both T cell lineages, the patients in the SLE-2 group expressed a stronger type I IFN signature. In CD4+ T cells, this was accompanied by genes involved in JAK-STAT signaling and T cell co-stimulation, whilst in CD8+ T cells the increased ISG expression was associated with genes involved in cell cycle, response to DNA damage and apoptosis (Fig. 1a, b). Gene set variation analysis (GSVA) and weighted gene co-expression analysis (WGCNA)[25,26] revealed that, in CD8+ T cells, genes belonging to mitochondria-induced apoptosis and DNA damage response pathways correlated with disease activity and they were co-expressed with the IFN signature that, on its own, did not associate with disease activity or other clinical criteria (Supplementary Fig. 2a). Intriguingly, the co-expression modules that showed the most significant correlation with disease activity included mtDNA-encoded genes involved in OXPHOS in both CD4+ and CD8+ T cells (Supplementary Fig. 2b, c), indicating a possible regulation of mitochondrial genes by type I IFNs.

Comparison of the transcriptomic profiles between SLE-1 and SLE-2 groups revealed that besides the ISGs the mtDNA-encoded OXPHOS genes were the most differentially expressed (Fig. 1c, d). Among the ~80 proteins constituting the ETC, 13 are encoded by genes located in the mitochondrial DNA, and the remaining are encoded by nuclear DNA (nDNA). Heatmap of all OXPHOS genes showed that SLE-2 patients downregulated predominantly mtDNA-encoded OXPHOS genes, but not nDNA-encoded genes (Supplementary Fig. 3a). This observation was validated by quantitative real-time (qPCR) (Supplementary Fig. 3b, c). The suppression of mitochondria-derived genes in SLE-2 group was more pronounced in CD8+ T cells than in CD4+ T cells (Supplementary Fig. 3b). Pathway analysis of genes involved in metabolic functions showed

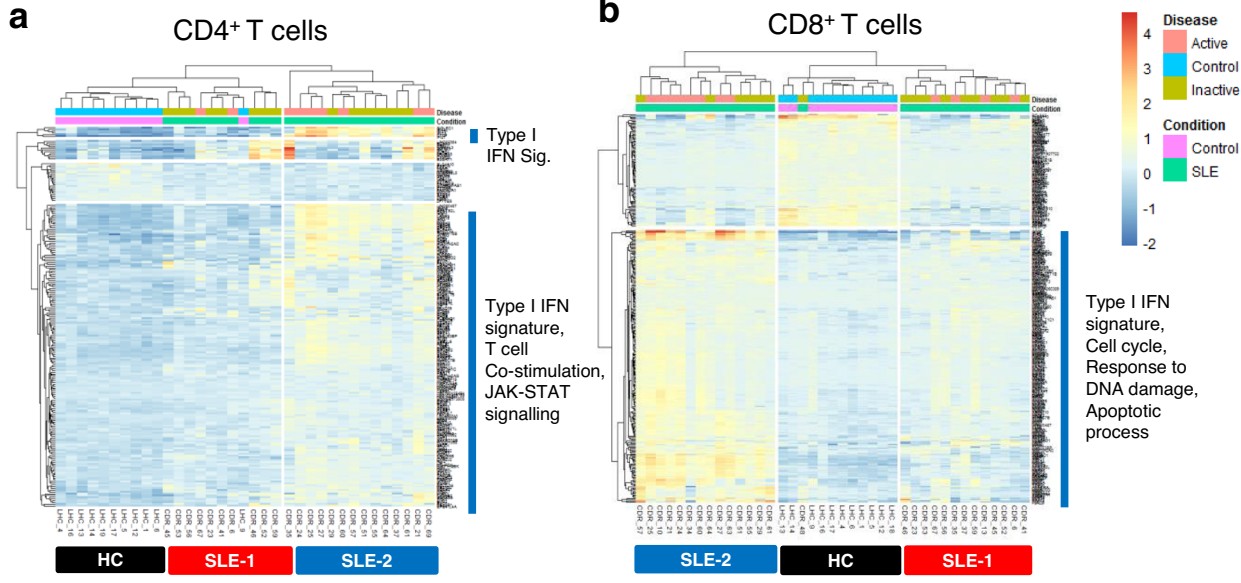

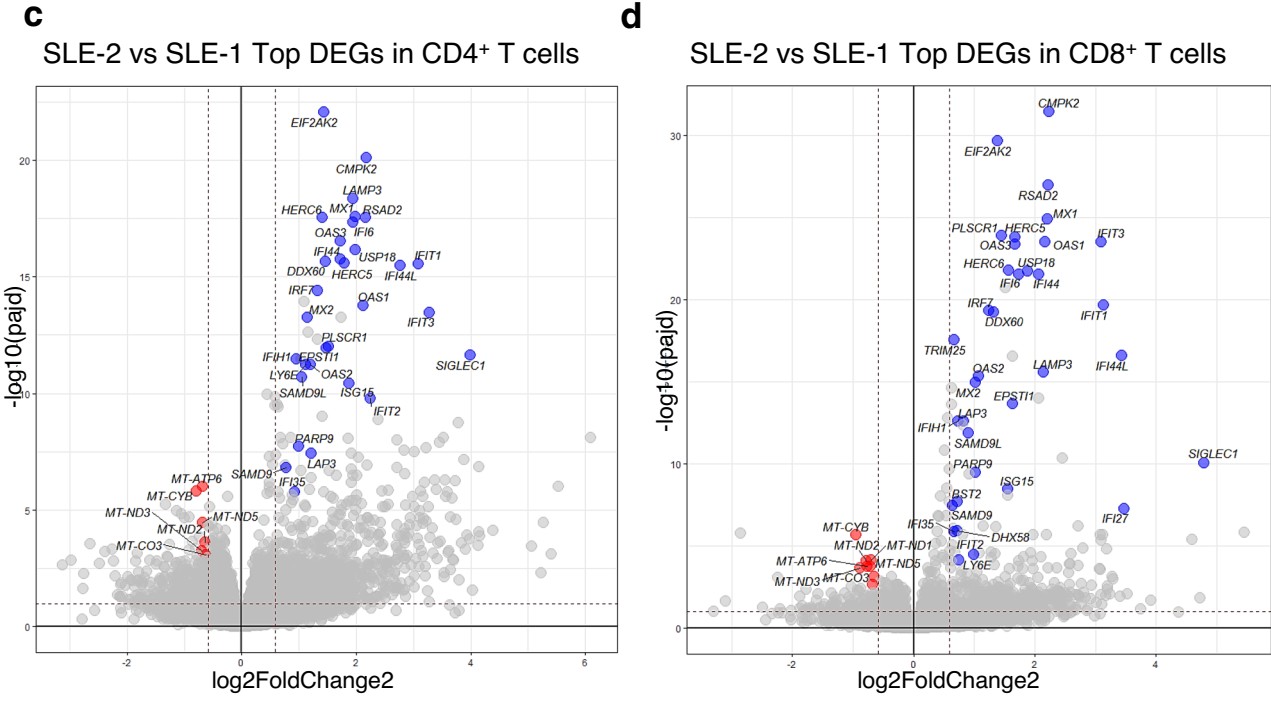

**Fig. 1 Transcriptomic analyses of CD8⁺ T and CD4⁺ T cells from healthy volunteers and patients with SLE. a, b** Unsupervised hierarchical clustering of all differentially expressed genes in CD4⁺ and CD8⁺ T cells between HC and SLE patients. **a** In CD4⁺ T cells, SLE-1 (active SLE $n = 2$, inactive SLE $n = 8$) and SLE-2 (active SLE $n = 7$, inactive SLE $n = 7$). **b** In CD8⁺ T cells, SLE-1 (active SLE $n = 4$, inactive SLE $n = 9$) and SLE-2 (active SLE $n = 8$, inactive SLE $n = 7$). HC healthy control. **c, d** Volcano plots from RNA-seq analysis showing ISGs and mtDNA-encoded oxidative phosphorylation (OXPHOS) genes as the most differentially expressed genes in CD4⁺ (**c**) and CD8⁺ T (**d**) cells between SLE-1 and SLE-2 groups. Blue dots indicate type I IFN-inducible genes and red dots indicate mitochondria-encoded OXPHOS genes.

significant downregulation of pathways involved in tricarboxylic acid (TCA) cycle, ETC, and ATP synthesis in SLE-2 compared to SLE-1 patients (Supplementary Fig. 3d).

Taken together, the transcriptomic analysis of purified T cells from SLE patients segregated them into two groups based on the level of ISG expression, and patients expressing a high type I IFN signature displayed reduced expression of mitochondria-encoded genes and mitochondria-associated metabolic pathways, which was

more pronounced in CD8⁺ T cells. This suggests a dysregulation of mitochondrial functions linked to type I IFNs in SLE, a hypothesis that we sought to test next.

**Aberrant mitochondria in CD8⁺ T cells from patients with high expression of ISGs.** Prior to studying the potential link between type I IFN signaling and mitochondrial abnormalities, we applied a combination of in silico and in vitro methods to

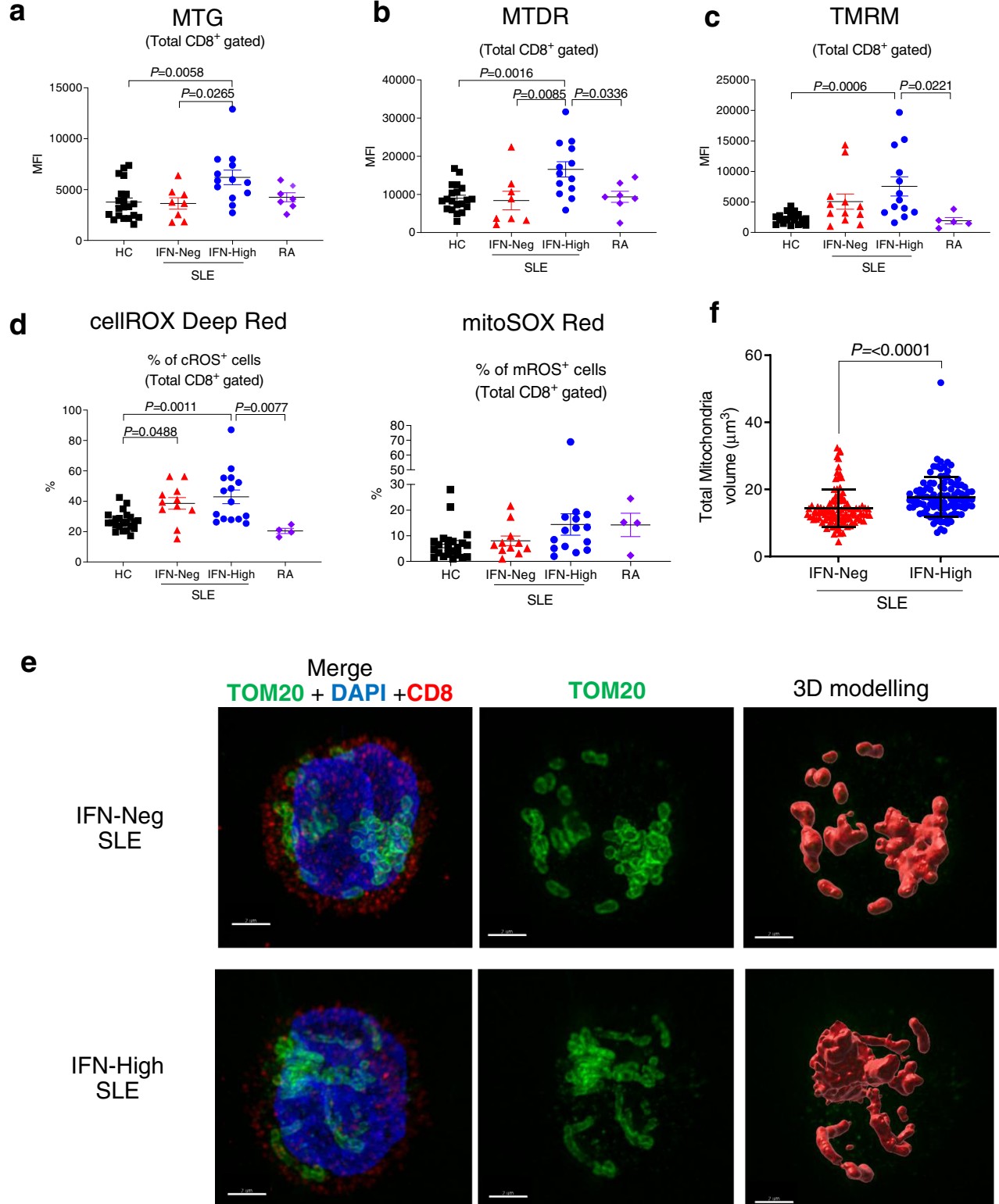

stratify the SLE patients according to the degree of the type I IFN signaling: high level of ISGs (IFN-High), low level of ISGs (IFN-Low), and no ISGs (IFN-Neg). By applying the method used by Kennedy et al.[27] with three type I IFN-induced genes (*HERC5*, *EPSTI1*, and *CMPK2*), we established a cut-off interferon score matrix (ISM) (IFN-High: ISM > 1.5; IFN-Low: ISM 0.5–1.5; IFN-Neg: ISM < 0.5) that highly correlated with our RNAseq-derived score (Supplementary Fig. 4).

Next, we investigated the mitochondrial phenotype in T cells from IFN-High and IFN-Neg SLE patients and included rheumatoid arthritis (RA) patients as disease controls (Supplementary Table 3). We found that CD8+ T cells from IFN-High SLE patients, compared to those from the other groups (HC, IFN-Neg SLE and RA patients), displayed increased mitochondrial size (Fig. 2a, b) and membrane hyperpolarization (Fig. 2c). The most pronounced mitochondrial changes were present in the effector memory

**Fig. 2 Mitochondrial changes in total CD8$^+$ T cells from patients with SLE.** Gated CD8$^+$ T cells were stained with **a** MitoTracker green (MTG) (HC $n =$ 20; IFN-Neg $n = 8$; IFN-High $n = 13$; RA $n = 7$); **b** membrane potential dependent-Mitotracker Deep Read (MTDR) (HC $n = 20$; IFN-Neg $n = 8$; IFN-High $n = 13$; RA $n = 7$); **c** Tetramethylrhodamine (TMRM). (HC $n = 22$; IFN-Neg $n = 12$; IFN-High $n = 13$; RA $n = 5$); Mean fluorescence intensity (MFI) data are shown. **d** Proportions of CD8$^+$ T cells positive for cROS (cellROX Deep Red) and mROS (MitoSOX Red) are shown. (HC $n = 22$; IFN-Neg $n = 12$; IFN-High $n = 13$; RA $n = 5$); **a–d** Data presented as mean ± S.E.M. Each symbol represents an individual. **e**, **f** Isolated CD8$^+$ T cells from IFN-Neg and IFN-High SLE patients were mounted on poly-L-lysine-coated coverslips, fixed and stained for TOM20 (green), CD8 (red) and DAPI (blue) as described in "Methods" section. Representative SIM images derived from maximal projection analysis (**e**) and total mitochondrial volume (**f**) of three independent samples per condition, each with at least 14 cells analysed (range 14–43 cells per sample). Scale bars, 2 mm. Data presented as mean ± S.E.M. **a–d** One-way ANOVA, **f** Two-tailed Mann–Whitney test; only significant differences are indicated. HC healthy controls, RA rheumatoid arthritis patients. Source data are for this figure provided as a Source Data file.

CD8$^+$ T cells, whilst central memory CD8$^+$ T cells showed no abnormalities (Supplementary Figs. 5a and 6a–c). The proportion of CD8$^+$ T cell subsets between IFN-High and IFN-Neg SLE patients was similar (Supplementary Fig. 6d). In the IFN-High SLE patients we also observed an increased fraction of CD8$^+$ T cells positive for cellular ROS (cROS), but only a trend in the proportion of mitochondrial ROS (mROS) stained cells (Fig. 2d and Supplementary Figs. 5b, c, and 6e). A higher percentage of cROS positive cells was also detected in the IFN-Neg SLE patients, suggesting a disease-specific effect (Fig. 2d). To visualise the morphological changes, we used super-resolution structured illumination microscopy (SR-SIM) with a combination of fluorescently labelled antibodies specific for a mitochondrial protein subunit of TOM20 (green) and CD8 (red) with DAPI (blue) for nucleus staining. The microscopic images complemented our flow cytometry data showing that in the CD8$^+$ T cells from IFN-High SLE patients the mitochondria were morphologically different compared to the ones in IFN-Neg SLE patients (Fig. 2e). Additionally, image analysis showed that the average total mitochondrial volume per cell was higher in IFN-High CD8$^+$ T cells (Fig. 2f). By scoring the images blindly we found that IFN-High SLE patients had a higher percentage of CD8$^+$ T cells with elongated mitochondria (21.7 ± 7.4) compared to the IFN-Neg SLE (9 ± 3.4). Of note, the analysis of CD4$^+$ T cells failed to detect any significant change in any of the mitochondrial parameters assessed (Supplementary Fig. 7). Despite the increased mitochondrial size and activity in CD8$^+$ T cells we found no differences in the mtDNA copy number (Supplementary Fig. 8a). Similarly, we did not detect any difference in the protein expression of mtDNA-encoded and nDNA-encoded OXPHOS genes (Supplementary Fig. 8b, c) in both T cells.

Taken together, the data indicate that in SLE patients the prolonged IFNα exposure may trigger mitochondrial changes in CD8$^+$, but not in CD4$^+$, T cells and this may lead to metabolically dysregulated cells with potential implications for their function.

**CD8$^+$ T cells from patients with high IFN signature have reduced SCR.** The mitochondrial changes detected in the CD8$^+$ T cells from IFN-High SLE patients prompted us to evaluate their metabolic functions. We next compared mitochondrial respiration and aerobic glycolysis in ex vivo CD4$^+$ and CD8$^+$ T cells sorted from peripheral blood of SLE patients and HC. Using the Seahorse extracellular flux analyzer we found that CD8$^+$ T cells from IFN-Neg patients had similar basal and maximal oxygen consumption rate (OCR) to the HC, whilst CD8$^+$ T cells from IFN-High patients showed less basal and maximal OCR (Fig. 3a). Importantly, in IFN-High CD8$^+$ T cells the spare respiratory capacity (SRC), that reflects the cell ability to adapt to increased energy demand, was significantly reduced (Fig. 3a). The decrease in SRC was accompanied by some reduction of intracellular ATP level (Supplementary Fig. 9a). On the other hand, no marked abnormalities in OCR or SRC were detected in the CD4$^+$ T cells isolated at the same time from the same patients (Fig. 3a). In

addition, glycolysis, measured by extracellular acidification rate (ECAR), was similar in all groups in both CD4$^+$ and CD8$^+$ T cells (Supplementary Fig. 9b).

We next explored whether the metabolic abnormalities observed ex vivo in the CD8$^+$ T cells from the IFN-High patients could result in T cell defects. We observed an increased spontaneous cell death in the CD8$^+$ T cells from IFN-High SLE patients compared to HC and IFN-Neg SLE patients (Fig. 3b). However, when the IFN-High CD8$^+$ T cells were cultured in vitro they lost their IFN signature very quickly and by day 3 the ISM was equivalent to the one observed in IFN-Neg cells (Supplementary Fig. 9c), a change that prevented us from pursuing further in vitro functional assays with the cells from the SLE patients.

In summary, the metabolic analysis revealed that the larger and hyperpolarized mitochondria in CD8$^+$ T cells from IFN-High SLE patients were bioenergetically impaired. Collectively, the data also suggest that the persistent activation of type I IFN pathways may reduce the metabolic fitness of CD8$^+$ T cells and thus their ability to survive.

**TCR and IFN signalling together induce CD8$^+$ T cell metabolic rewiring.** In order to investigate the sequential events leading to the mitochondrial and metabolic changes seen in IFN-High CD8$^+$ T cells, we next used cells from HC and a combination of prolonged IFN treatment and TCR activation, two essential cellular signals present in SLE patients. We initially cultured peripheral blood mononuclear cell (PBMCs) from HC with varying concentrations of IFNα (10, 100, and 1000 U/ml). The 1000U/ml dose of IFNα increased ISG expression to the same degree seen in IFN-High SLE patients (Supplementary Fig. 10a) and thus this dose was used in all subsequent experiments. Importantly, while exposure of CD8$^+$ T cells to IFNα for 2 days did not trigger any significant mitochondrial or metabolic changes (Supplementary Fig. 10b–f), 7-day IFNα exposure, especially combined with T cell activation, induced downregulation of the mtDNA-encoded gene expression (Fig. 4a) and changes in the mitochondria (Fig. 4b), that were similar to the ones observed in the CD8$^+$ T cells from IFN-High SLE patients (Fig. 2a, b). We then analysed the oxidative capacities of the CD8$^+$ T cells and found that 7-day IFNα stimulation with or without CD3/CD28 activation significantly enhanced the basal OCR, but not the maximal OCR, resulting in a decreased SRC when the value was normalized to basal level of each sample, especially when the IFNα exposure was combined with T cell activation (Fig. 4c). Consistent with the ex vivo patient data (Supplementary Fig. 9b), we found that the IFNα priming did not alter the basal ECAR (Supplementary Fig. 11a).

The reduced SRC capacity in the CD8$^+$ T cells after prolonged stimulation with IFNα and anti-CD3/CD28 beads suggests impaired bioenergetic fitness. To corroborate this, we analysed their glycolytic and oxidative response upon restimulation with anti-CD3/CD28 beads injected during the

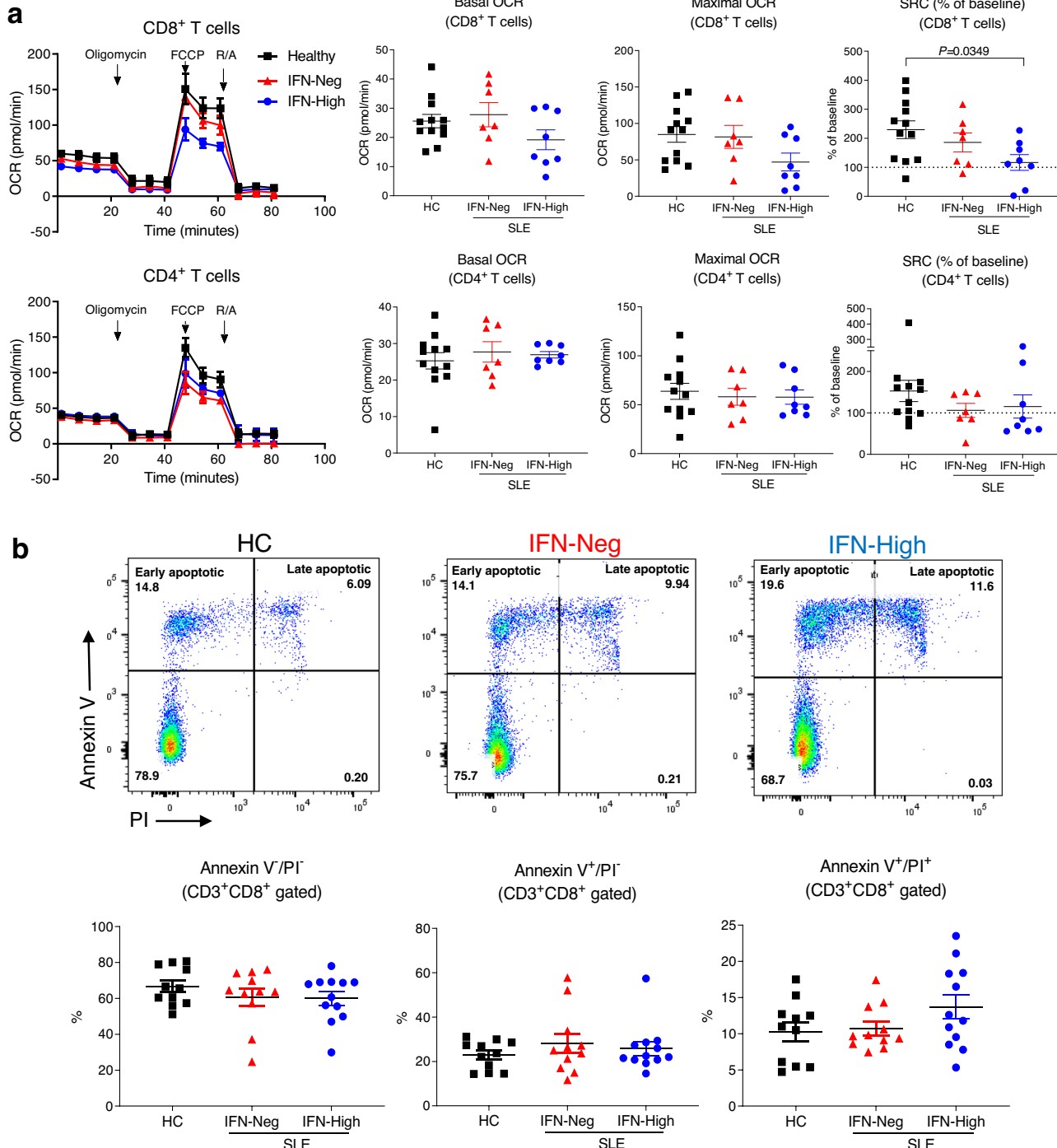

**Fig. 3 Metabolic analysis and apoptotic rate in CD8+ T cells from patients with SLE. a** Representative oxygen consumption rate (OCR) (left panels) and summary graphs (right panels) of sorted CD4+ and CD8+ T cells from controls, IFN-Neg and IFN-High SLE patients during a mitochondrial stress test. Spare respiratory capacity (SRC) was normalized to basal level of each individual. Oligomycin, carbonylcyanide p-trifluoromethoxyphenylhydrazone (FCCP), and rotenone/antimycin A (R/A) were added to the cells as indicated. Each symbol represents an individual (HC $n = 12$; IFN-Neg $n = 7$; IFN-High $n = 8$). **b** PBMCs from IFN-Neg and IFN-High SLE patients and HC were rested in culture for 48 h followed by staining with Annexin V and propidium iodide (PI). Representative flow cytometry plots of CD3+ CD8+ gated cells and summary graphs. Each symbol represents an individual (HC $n = 11$; IFN-Neg $n = 11$; IFN-High $n = 12$) **a, b** Data presented as mean ± S.E.M. One-way ANOVA, only significant differences are indicated, HC healthy controls. Source data for this figure are provided as a Source Data file.

extracellular flux assay. Under these experimental conditions the IFNα-stimulated CD8+ T cells showed, after the initial burst, a more rapid decline in their oxidative and glycolytic capacities compared to the CD8+ T cells stimulated with CD3/CD28 only. Similar results were also observed upon restimulation with PMA/ionomycin used as non-specific stimuli (Fig. 4d and Supplementary Fig 11b).

In summary, the in vitro studies using CD8+ T cells from HC demonstrated that whilst a prolonged IFNα treatment alone was able to trigger detectable metabolic rewiring, only the combined

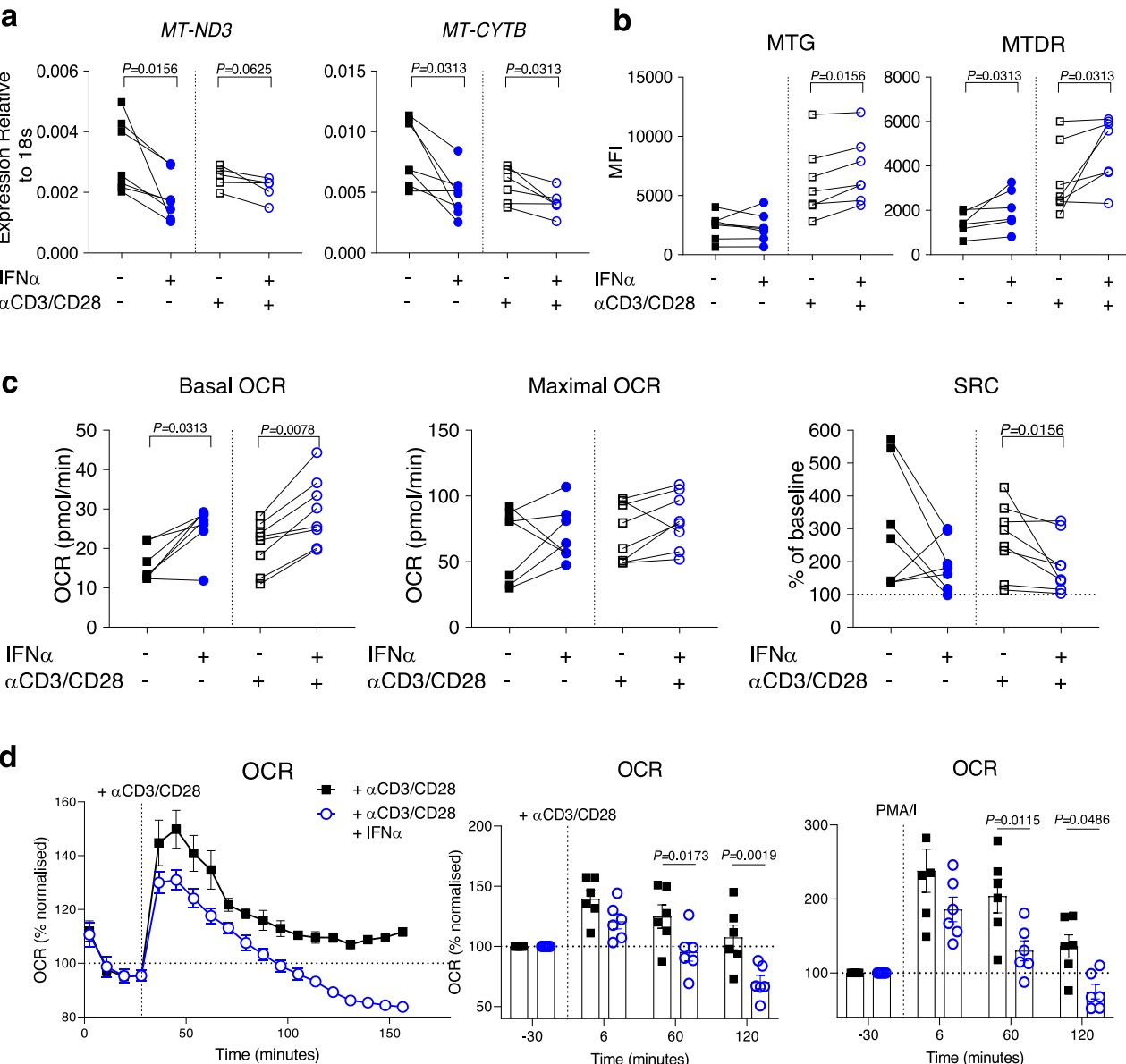

**Fig. 4 Metabolic changes triggered by 7-day IFNα exposure and T cell activation. a**, **b** Purified CD8+ T cells from healthy donors treated with or without αCD3/CD28 beads in the presence or absence of 1000U/ml IFNα for 7 days. Mitochondria-encoded gene expression ($n = 5-7$) (**a**) and changes in mitotracker stainings ($n = 6-7$) (**b**) are shown. **c**, **d** PBMCs from healthy donors treated with or without αCD3/CD28 beads in the presence or absence of 1000U/ml IFNα for 7 days. CD8+ T cells were FACs-sorted and analysed using the extracellular flux assay. **c** Graphs showing the basal and maximal OCR levels under the different experimental conditions as indicated. Spare respiratory capacity (SRC) was normalized to basal level of each individual ($n = 7-8$). **d** Oxidative response of FACs-sorted CD8+ T cells upon re-stimulation with anti-CD3/CD28 beads or PMA/I injected during the extracellular flux assay. A representative graph (left panel) and levels (right panels) at different time points normalised to basal level of each individual are shown ($n = 6$). **a**–**d** Data presented as mean ± S.E.M. Each symbol represents one donor. **a**–**c** Two-tailed Wilcoxon matched-pairs signed rank test was utilized, **d** two-way ANOVA; only significant differences are indicated; MFI mean fluorescence intensity, MTG MitoTracker green, MTDR membrane potential dependent-Mitotracker Deep Read, MT-ND3 mitochondrially encoded NADH:Ubiquinone oxidoreductase core subunit 3, MT-CYTB mitochondrially encoded cytochrome B, PMA/I phorbol12-myristate13-acetate/Ionomycin. Source data for this figure are provided as a Source Data file.

exposure of IFNα and T cell activation could phenocopy the mitochondrial and metabolic abnormalities observed in CD8+ T cells from the IFN-High SLE patients.

**IFNα exposure induces cell death in chronically activated CD8+ T cells.** Having established experimental conditions that recreated the metabolic changes seen in SLE CD8+ T cells, we next investigated the downstream effects. Consistent with the ex vivo data from the SLE patients, we found that prolonged IFNα exposure combined with T cell activation promoted spontaneous death of

CD8+ T cells (Fig. 5a). In agreement with the spontaneous cell death data, we found that, upon rechallenge, the percentage of proliferating CD8+ T cells expressing Annexin V was significantly increased in the cells exposed to IFNα (Fig. 5b). In addition, we detected fewer degranulated (CD107a positive) and activated (CD69 and CD25 positive) CD8+ T cells among the IFN-stimulated cells (Fig. 5c). These changes were detected only at 4 h after restimulation (Supplementary Fig. 12a), and were accompanied by a decrease in the secretion of TNFα and a trend of reducing secretion of granzyme B and IFNγ (Supplementary Fig. 12b).

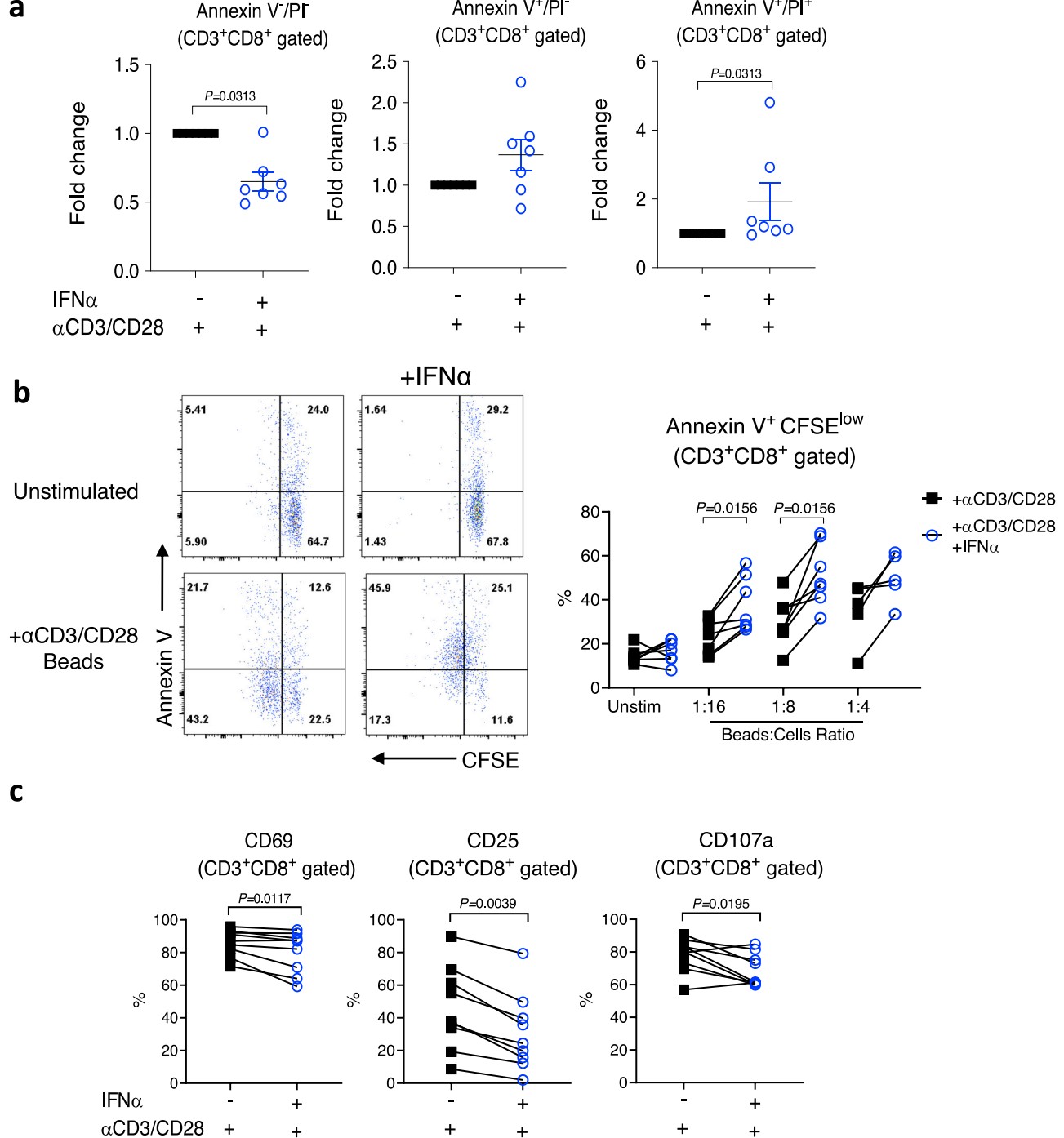

**Fig. 5 Changes induced by 7-day exposure to IFNα and T cell activation.** PBMCs from HC were stimulated with αCD3/CD28 beads in the presence or absence of 1000U/ml of IFNα for 7 days. **a** αCD3/CD28 beads were removed from the culture and cells were left in culture for additional 48 h. Annexin V and propidium iodide (PI) staining of gated CD3+CD8+ cells was used to detect spontaneous cell death. Data shown are mean ± S.E.M of fold-change of each individual sample exposed to αCD3/CD28 beads only. Each dot represents one donor (*n* = 7). **b, c** αCD3/CD28 beads were removed from the culture and cells were rested overnight before labelling with CFSE and restimulation with αCD3/CD28 beads. **b** Representative flow plots of Annexin V staining and CFSE dilution (CD3+CD8+gated cells) at day 3 after re-stimulation (beads to cell ratio indicated). Percentages of Annexin V+CFSElowCD8+ T cells (*n* = 5–7). **c** Percentages of CD8+ T cells positive for CD69, CD25, and CD107a at 4 h of post restimulation (beads to cell ratio = 1:2). Each symbol represents one donor (*n* = 9). **a–c** Two-tailed Wilcoxon matched-pairs signed rank test; only significant differences are indicated; HC healthy controls. Source data for this figure are provided as a Source Data file.

Together the data suggest that the type I IFN signature altered the metabolism of CD8+ T cells in SLE patients resulting in reduced cell survival upon TCR rechallenge.

**IFNα exposure increases NAD+ consumption in CD8+ T cells.** To understand the mechanistic link between chronic type I IFNs and downstream mitochondrial and metabolic changes observed

in the CD8$^+$ T cells, we first explored which pathways correlated with type I IFN signaling in the transcriptomic data from the CD8$^+$ T cells of the SLE patients. We found that the nicotinate/nicotinamide metabolic pathway showed the highest correlation ($r^2 = 0.6562$; $P \leq 0.0001$, Pearson's correlation coefficient) (Fig. 6a) and NAD-consuming enzymes, such as the poly(ADP-ribose) polymerases (*PARP9, PARP10,* and *PARP12*) and *CD38*[28], were markedly upregulated in CD8$^+$ T cells from IFN-High SLE patients (Fig. 6b). Furthermore, TCR-stimulated CD8$^+$ T cells from healthy controls showed increased surface expression of

CD38 (Fig. 6c) and induced gene expression of *PARP9* and *PARP10* after IFNα treatment (Supplementary Fig. 13a), suggesting a direct effect of IFNα on the nicotinate/nicotinamide metabolic pathway. Based on these data, we reasoned that the upregulation of the expression of NAD-consuming enzymes elicited by type I IFN signaling in activated CD8$^+$ T cells could lead to NAD+ depletion and subsequent mitochondrial dysfunction. Consistent with this hypothesis, we observed a significantly decreased NAD+/NADH ratio in CD8$^+$ T cells after prolonged IFNα and TCR stimulation (Fig. 6d), the experimental conditions

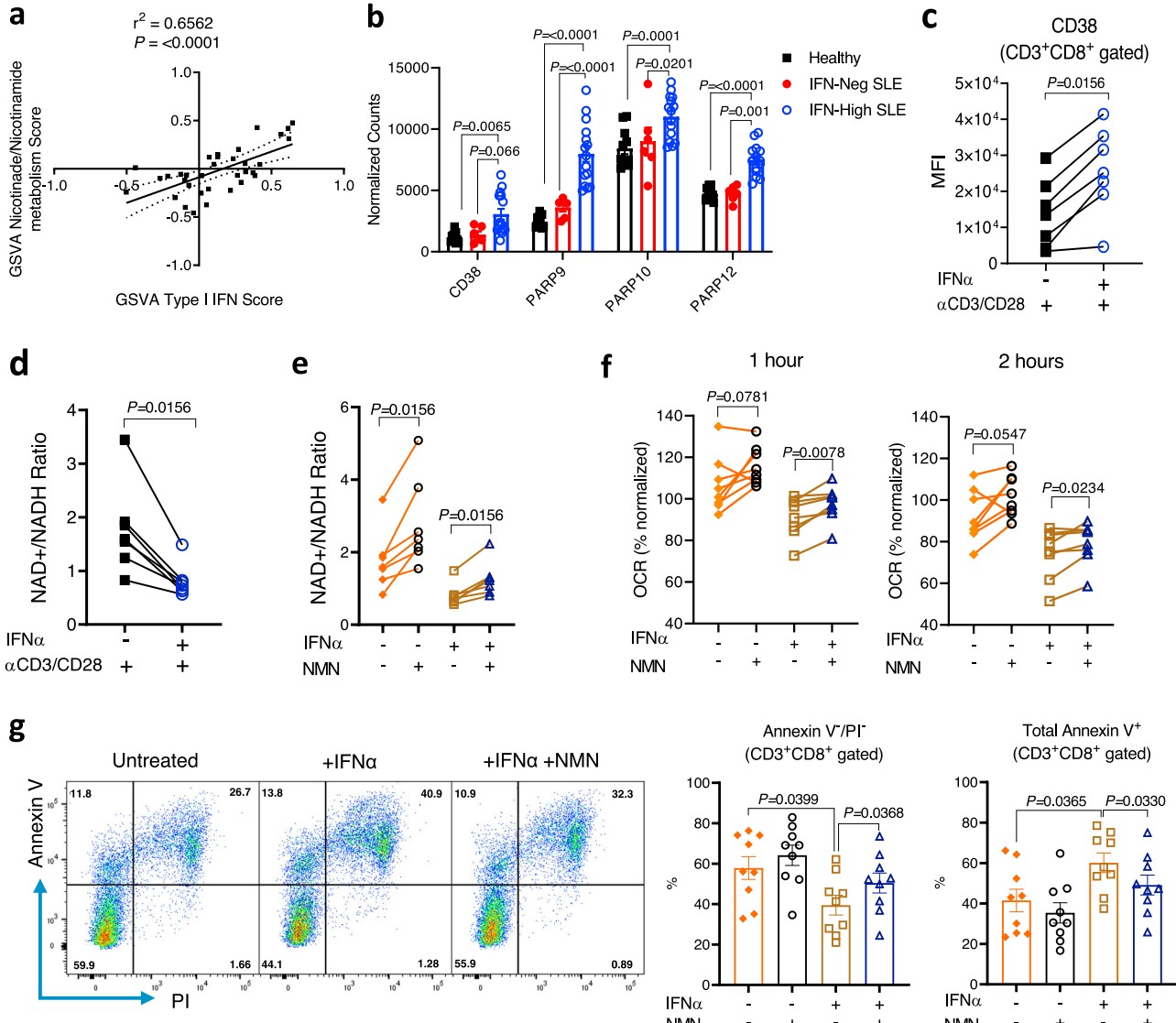

**Fig. 6 IFNα increases NAD+ consumption leading to reduced SCR and cell viability. a** Correlation between enrichment score for KEGG's NAD metabolic pathway and Type I IFN signalling using GSVA package. Data obtained from the transcriptomic analysis of the SLE cohort. Pearson correlation coefficient ($r^2$) and *P*-value are shown ($n = 34$). **b** Gene expression of NAD-consuming enzymes (CD38; PARP9, PARP10, and PARP12) in CD8$^+$ T cells from IFN-Neg ($n = 6$), IFN-High ($n = 15$) SLE patients, and HC ($n = 11$). Normalized read counts are shown. **c, d** Purified CD8$^+$ T cells from HC were treated with IL-2 (10U/ml) and αCD3/CD28 with or without 1000U/ml IFNα for 7 days. **c** CD38 expression was measured using flow cytometry. MFI is shown ($n = 7$). **d** NAD+/NADH ratio measured using NAD+/NADH Assay Kit (Abcam) ($n = 7$). **e–g** Purified CD8$^+$ T cells were stimulated as in **c** with or without addition of 1 mM NMN from day 3 to day 7. At day 7 CD8$^+$ T cells were isolated and rested overnight. **e** NAD+/NADH ratio ($n = 7$). **f** OCR levels 1 and 2 h after restimulation with αCD3/CD28 beads injected during the extracellular flux assay under the different experimental conditions as indicated. Data normalized to basal level of each individual ($n = 8$). **g** Representative flow plots (left panels) of Annexin V and PI staining (CD3$^+$CD8$^+$gated cells) after αCD3/CD28 restimulation for 3 days are shown. Percentage of live cells (Annexin V$^-$/PI$^-$) and apoptotic cells (Annexin V$^+$) under the different experimental conditions (right panels) are shown ($n = 9$). **a–g** Data presented as mean ± S.E.M. Each symbol represents one donor. **b** Two-way ANOVA; **c–f** Two-tailed Wilcoxon matched-pairs signed rank test was utilized; **g** Matched-pairs One-Way ANOVA; Only significant data are indicated; MFI mean fluorescence intensity, PARP poly(ADP-ribose) polymerases, NAD nicotinamide adenine dinucleotide, NMN β-Nicotinamide mononucleotide, OCR oxygen consumption ratio, PI propidium iodide. Source data for this figure are provided as a Source Data file.

mimicking the mitochondrial and metabolic abnormalities observed in CD8[+] T cells from IFN-High SLE patients.

To test whether a NAD+ supplementation could rectify the downstream effects of IFNα stimulation, we cultured the CD8[+] T cells with 1 mM β-Nicotinamide mononucleotide (NMN) to restore the NAD+/NADH ratio (Fig. 6e). We then assessed the oxidative response and cell viability upon restimulation with anti-CD3/CD28 beads. The NAD+ supplementation with NMN increased the OCR (Fig. 6f), but not the ECAR (Supplementary Fig. 13c), indicating increased mitochondrial capacity. Similar results were also observed upon restimulation with PMA/ionomycin used as a positive control (Supplementary Fig. 13d). Importantly, restoring NAD+ availability with the NMN treatment improved cell viability following prolonged IFNα and TCR stimulation (Fig. 6g) upon rechallenges, indicating that the NAD pathway modulated mitochondrial fitness and cell survival in these activated human CD8[+] T cells. Given that NAD+ supplementation has been reported to sustain mitochondrial fitness by reducing mROS levels in CD8[+] T cells[29], we explored this in our in vitro model. We found that IFNα exposure increased the percentage of activated CD8[+] T cells expressing mROS upon restimulation, and the NMN treatment was capable of attenuating this effect (Supplementary Fig. 13e).

In summary, our data indicate that prolonged IFNα exposure increased the consumption of NAD+ and reduced the NAD+/NADH ratio in human TCR-stimulated CD8[+] T cells. This led to defective mitochondrial respiration upon restimulation and reduced cell viability. The NAD+ precursor NMN rectified these metabolic changes and improved CD8[+] T cell viability.

## Discussion

It is recognized that most SLE patients have an increased expression of ISGs, but the biological and clinical implications of this remains unclear. Our T cell transcriptomic and metabolic studies in a clinically well-defined group of SLE patients with biopsy-proven nephritis showed that, high IFN signaling can drive changes in the mitochondrial metabolic pathways of CD8[+] T cells making them bioenergetically unfit and more prone to die. Using a combination of in silico and in vitro methods, here, we demonstrated that the metabolic rewiring observed in CD8[+] T cells from SLE patients was the result of prolonged IFNα exposure and TCR stimulation. Under the persistent combination of these two stimuli, a condition probably unique to SLE patients, the CD8[+] T cells displayed increased *PARP* gene expression associated with enhanced NAD+ consumption, reduced mitochondrial oxidative capacity and decreased survival (Fig. 7). Collectively, our findings demonstrated a link between high ISG signature and the metabolic fitness of SLE CD8[+] T cells, and their ability to survive under stress or in response to antigen rechallenges.

Over the past decade several transcriptomic studies have been conducted in SLE cohorts[30]. However, most of the studies have been conducted using whole blood and SLE patients with different clinical manifestations, an obstacle for inferring biological pathways beyond the shared IFN response signature. A recent integrated and multi-cohort analysis of 40 independent studies identifies a unique SLE MetaSignature of 93 genes in the blood and 85% of these are IFN-related[31]. Of note, in this study, the non-IFN SLE MetaSignature has a higher correlation with SLE disease activity than the IFN SLE MetaSignature. Consistent with this and other reports[4,27,32], our transcriptomic analysis of T cells from SLE patients with renal involvement found that ISG levels did not correlate with disease activity. Instead, we showed that a high type I IFN signature was associated with several mitochondrial abnormalities (e.g., decreased expression of mitochondria-encoded OXPHOS genes and larger and hyperpolarized mitochondria) in CD8[+] T cells. These abnormalities

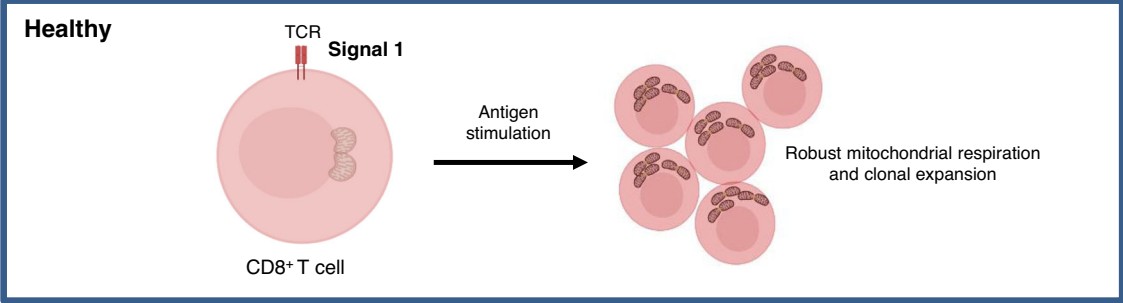

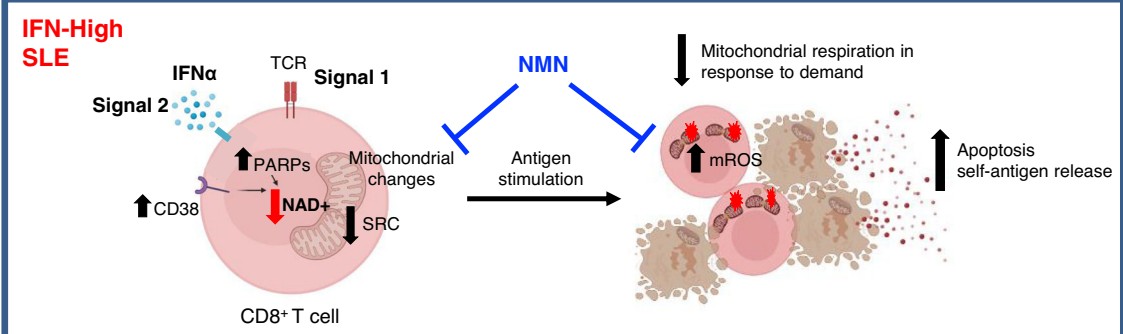

**Fig. 7 Type I interferons modulate the metabolic fitness of CD8[+] T cells in patients with SLE.** In HC, TCR signaling (signal 1) is essential for CD8[+] T cell survival. In IFN-High SLE patients, persistent activation of the TCR (signal 1) and the type I IFNα pathways (signal 2) triggers mitochondrial changes via increasing NAD+ consumption in CD8[+] T cells resulting in lower spare respiratory capacity (SRC) and thus decreased bioenergetic fitness. Upon increased energy demand such as in response to antigen challenge or stress, IFNα-stimulated CD8[+] T cells are more prone to die, and this could perpetuate autoimmunity by increasing the autoantigen load. NAD+ supplementation with NMN restored the NAD+ pool, increased the mitochondrial respiration, decreased mROS, and improved cell viability upon TCR restimulation. Figure was created with BioRender.com.

created functionally less capable mitochondria, resulting in increased cell death that in itself has been linked to SLE pathogenesis. Previous studies have reported a higher degree of apoptosis in T cells from SLE patients and shown a correlation with disease activity, however, no mechanistic link to the IFN signature was made[33].

One of the most interesting observations from our comparative transcriptomic analysis of purified CD4+ and CD8+ T cells was the relatively stronger association between gene expression signature and clinical parameters in CD8+ T cells, indicating that these cells may play an integral role in the disease pathogenesis. Consistent with this, a previous transcriptomic study using microarrays postulated that gene signatures from CD8+ but not CD4+ T cells, could predict long-term prognosis in several autoimmune conditions including SLE[34]. However, both protective and deleterious roles of CD8+ T cells have been reported[35,36], which may reflect distinct roles of these cells at different stages of SLE. In vitro stimulation of antigen-experienced antiviral CD8+ T cells from SLE patients shows less degranulation and reduced cytokine production, suggesting dysfunctional CD8+ T cell responses[37]. In contrast, other studies report increased frequency of circulating cytolytic CD8+ effector T cells in active SLE patients compared to healthy controls[35]. Tissue injury in SLE can be amplified by CD8+ T cells[38] and proteolysis by granzyme B, the serine protease released by cytotoxic CD8+ T cells, can generate unique autoantigen fragments contributing to increased autoantigen load[35]. Additionally, here we demonstrated that CD8+ T cells from IFN-High patients displayed increased spontaneous apoptotic rate, and upon restimulation they were more prone to cell death, which could exacerbate on-going autoimmunity by increasing exposure to self-antigens (Fig. 7). Besides the auto-antigen load, CD8+ T cells may also contribute to perpetuation of autoimmunity by regulating auto-antibody production as a distinct subset of PD1+CXCR5+ CD8+ T cells is recently identified within the B cell follicle and germinal center[39]. These cells are shown to have similar function to CD4+ T follicular helper cells supporting auto-antibody production. Together all these studies suggest unprecedent roles in autoimmunity for CD8+ T cells, which warrant further investigations into the immunomodulatory functions of CD8+ T cells in SLE pathogenesis.

There is accumulating evidence that while acute type I IFN signaling improves immune responses prolonged stimulation leads to regulatory pathways, that limit immunity and promote chronic disease[40]. During chronic viral infections, bystander CD8+ T cells exposed to IFNα are less able to proliferate upon stimulation[41] and blockade of type I IFN signaling enhances virus clearance[42]. However, these regulatory pathways are not clearly defined and are cell type and context-dependent. Our comparison between IFN-Neg and IFN-High SLE T cells offers unique insights into the impact of chronic type I IFN signaling in SLE. Consistent with previous transcriptomic studies of SLE CD4+ T cells, we found that ISG signature in these cells was associated with JAK-STAT, T cell co-stimulation pathways and tissue homing signatures[24] (Fig. 1 and Supplementary Fig. 1), indicating that chronic type I IFN signaling may promote CD4+ T cell response. However, chronic type I IFN signaling appears to be detrimental to CD8+ T cells as we found that DNA damage, apoptosis signatures, and the NAD pathway correlated with ISGs. Consistent with this, metabolic analyses showed that CD8+ T cells from IFN-High SLE patients or after prolonged exposure to IFNα in vitro were metabolically impaired and less capable to survive. Collectively, our data suggest the modulation of metabolism as one of the regulatory mechanisms in CD8+ T cells restraining their immune response to persistent type I IFN signaling.

Metabolic adaptation is an important feature of CD4+ and CD8+ T cell functions[43]. Following antigen recognition, mitochondria oxidation and aerobic glycolysis work in concert to promote clonal expansion, effector functions, and survival of T cells. The imbalance between these two metabolic pathways

results in aberrant T cell responses and reduced survival. Abnormal mitochondrial metabolism in SLE has been primarily described in CD4+ T cells. Such defects include mitochondrial hyperpolarization, enlarged mitochondrial size, ATP depletion, and increased ROS production[44]. Increased OXPHOS and glycolysis have also been reported in CD4+ T cells from both SLE patients and lupus prone mice[20]. In our study, we could not replicate these observations and this may be related to the different clinical cohorts. Furthermore, for the functional studies we selected patients who were clinically stable and did not require change in treatment. Of note, in all previous studies the link between the mitochondrial defects and ISGs was not explored. Indeed, very few studies have investigated the effects of type I IFNs on T cell metabolism, particularly in the context of chronic IFNα exposure. A study in mice shows that in memory-like OT-1 CD8+ T cells, IFNα induces an increase in fatty acid oxidation driven OCR[14]. Other studies have shown that IFNα treatment in human B and T cell lines can cause a reduction in gene expression of mitochondria-derived genes[13,45]. However, these in vitro studies investigate relatively short-term exposure to IFNα and we found that type I IFN exposure up to 48 h had no detectable metabolic effects on human primary T cells. In contrast, longer (up to 7 days) exposure reduced the SRC in CD8+ T cells, but not CD4+ T cells. Our data, therefore, reinforce the notion that immunometabolism plays an important role in SLE pathogenesis, and highlight that immune cells use different metabolic pathways in response to acute or chronic changes. Further studies will be required to explore the mechanisms regulating these different metabolic programs.

Defects in mitophagy in cells from SLE patients have been described to be associated with IFN signature[12,46]. In addition, the formation of megamitochondria in PBMCs from SLE patients is associated with increased mitochondrial fusion via defects of mitochondrial fission initiator, Drp1 protein[47]. Indeed, we observed enlarged mitochondria in the CD8+ T cells from IFN-High SLE patients (Fig. 2e) that may suggest enhanced mitochondrial fusion or defects in clearance through mitophagy. However, our transcriptomic analysis did not reveal gene signatures associated with mitophagy (such as *PINK1*, *PRKN*, *BNIP3L*, *BNIP3*, *FUNDC1*, *ATG7* genes) to be differentially expressed in IFN-High SLE patients. Instead, we showed that type I IFN signaling impaired mitochondrial metabolism in TCR-stimulated CD8+ T cells via modulation of the NAD metabolic pathway. NAD+ is known to be a metabolic cofactor with critical roles in the maintenance of mitochondrial fitness[48]. NAD+ can affect mitochondrial function through regulation of mitochondrial unfolded protein response, or via providing electrons to respiratory chain Complex I. NAD+ is also crucial to maintain cellular redox homeostasis through the generation of NADPH.

Our study showed that the expression of *CD38* and *PARP* genes, well-known NAD-consuming enzymes, was associated with the type I IFN signature in SLE patients (Fig. 6a) and was inducible by IFNα exposure in vitro (Fig. 6b). Indeed, a recent study has linked the increased expression of CD38 in CD8+ T cells to decreased effector functions in SLE patients with infections[49]. However, the correlation with type I IFN signature was not investigated. In cancer, a crucial role of the CD38-NAD+ axis in T cells has also been suggested where reduced CD38 expression has been shown to correlate with higher cellular NAD+ levels, enhanced oxidative phosphorylation and glutaminolysis and improved T cell effector functions[50]. Activation of PARP proteins is known to be induced by DNA damage[51]. Indeed, deficiency in DNA damage response has been described in SLE and other auto-immune diseases. PBMCs from SLE patients showed decreased PARP activity upon UV-irradiation compared to healthy controls[52]. We observed mROS accumulation after IFN treatment, and this could have induced

DNA damage and triggered the increased PARP expression. Of note, while ISGs alone were not associated with disease activity, genes belonging to mitochondria-induced apoptosis and DNA damage response pathways significantly correlated with disease activity (Supplementary Fig. 2a) indicating the potential contribution of these pathways to the disease pathogenesis. A recent study reported that upon DNA damage, cells became more dependent on OXPHOS than glycolysis as inhibition of OXPHOS, but not glycolysis, induce ATP deprivation and cell death[53]. Thus, the increased PARP expression associated with the increased NAD+ consumption in response to DNA damage could have triggered the metabolic shift in the CD8+ T cells from IFN-High SLE patients, a change critical for their survival. Moreover, it has been reported that NAD+ supplementation can restore mitochondrial function and can reduce accumulation of mROS in CD8+ T cells[29]. Consistent with this, our data in human activated CD8+ T cells showed that NAD+ supplementation with NMN restored the NAD+/NADH ratio, increased the mitochondrial respiration, decreased mROS and, more importantly, improved cell survival in response to antigen re-challenges. Thus, our study uncovered a link between IFNα exposure and the NAD metabolic pathway and the critical role of NAD+ pathway in the metabolic fitness of TCR-stimulated CD8+ T cells. Whether boosting NAD+ levels could have beneficial effect in IFN-High SLE patients will require further investigation.

Our findings linking the type I IFN signature to dysregulation of the mitochondrial metabolism in CD8+ T cells go beyond the existing knowledge about the role of type I IFNs in SLE. We show the regulation of NAD pathway to be one of the mechanisms by which type I IFNs affect mitochondrial functions in SLE. However, other mechanisms are also likely to contribute to the metabolic impairment of the CD8+ T cells from IFN-High SLE and further studies are warranted to explore these additional potential pathways.

Although IFN signature cannot be used as a disease biomarker for SLE[4], in our SLE cohort as well in other cross-sectional studies, IFN-High patients tend to have a more severe disease course[54], suggesting that ISGs do play a role in the disease pathogenesis. In addition, a recent phase III clinical trial (TULIP 2, NCT02446899) with a human monoclonal antibody (Anifrolumab) that specifically blocks IFNAR1 (type I interferon receptor subunit 1) meet the primary end-points and thus this new therapeutic approach promises to change the standard of care for SLE patients[55]. The trial data also suggest a better response in IFN-High SLE patients. Consistent with this, our study linking type I IFNs to CD8+ T metabolic fitness and survival advocates that the underlying disease mechanisms between IFN-High and IFN-Neg SLE patients may be different and thus patient stratification based on IFN signature should be considered for treating SLE patients. In addition, our data support the concept of exploring metabolic pathways as therapeutic targets in SLE.

## Methods

**Patients.** All patients with SLE in the study met the revised American College of Rheumatology criteria[56] and the SLICC[57] criteria and had biopsy-proven nephritis. Lupus nephritis (LN) subsets were categorized according to the International Society of Nephrology/Renal Pathology Society classification. Patients who had received cyclophosphamide and/or B-cell depletion within 6 months were excluded. For the transcriptomic study 29 patients with SLE were recruited from the Imperial Lupus Center and the details are presented in Supplementary Table 1. The Systemic Lupus Erythematosus Disease Activity Index (SLEDAI score) and British Isles Lupus Assessment Group (BILAG) index were used for clinical classification; active disease was defined as a SLEDAI > 6 and inactive SLEDAI < 4. Active renal disease was defined as urine protein:creatinine ratio >50 mg/mmol together with biopsy-proven class III or IV or V LN within 3 months of recruitment. Inactive renal disease was defined as patients with a history of biopsy-proven LN, on a prednisolone dose of ⩽10 mg daily together with renal BILAG domain grade D, protein:creatinine ratio <50 mg/mmol and no change in SLE-related medication within 12 months before recruitment. Eleven healthy female volunteers (with no family history of autoimmune disease) served as age-matched and ethnicity-matched controls. The details of the SLE cohort and healthy volunteers involved in

the metabolic and functional studies are summarised in Supplementary Table 3. Seven female patients with RA were recruited as disease controls for the metabolic study. All patients gave informed consent and samples were collected as a sub-collection registered with the Imperial College Healthcare Tissue Bank (licence: 12275; National Research Ethics Service approval 17/WA/0161). The Tissue Management Committee of the Imperial College Healthcare Tissue Bank approved the study (ref: R13010a).

**PBMC separation and T cell isolation.** PBMCs were obtained by density gradient centrifugation using Lymphoprep (STEMCELL Technologies, Canada). In short, approximately 50 ml of blood were diluted 1× with phosphate buffered saline (PBS) with addition of 2% FBS and layered on top of 15 ml of Lymphoprep solution. The sample was then centrifuged at 800 × g for 20 min at room temperature without break. PBMCs were collected from the interface and washed twice with PBS/2% FBS. To maximize the purity of CD4+ and CD8+ T cells, T cells were first enriched using negative magnetic selection Pan T Cell Isolation Kit (Miltenyi Biotec) per manufacturer's instruction. For metabolic studies, as indicated, the negatively selected T cells were then labelled with anti-CD4 and anti-CD8 antibodies and sorted using Aria II FACS (Becton-Dickson). For RNA sequencing experiments, CD4+ and CD8+ T cells were positively isolated using Dynabeads FlowComp Human CD4 or CD8 Kit (Invitrogen) according to the manufacturer's protocol. The average purity as assessed by flow cytometry was more than 95%.

For some in vitro culture experiments CD8+ T cells were isolated from buffy coat cones (NHSBT) from healthy donors. CD8+ T cells were purified using human CD8+ T Cell Isolation Kit (Myltenyi Biotec) according to the manufacturer's protocol. Cells were centrifuged and resuspended in complete medium (CM) containing RPMI 1640, 10% of heat-inactivated FBS, 1% of L-glutamine with penicillin-streptomycin, 0.1% of 50 mM β-mercaptoethanol, 2% of 1 M HEPES buffer solution, 1% of MEM non-essential amino acids solution and 2% of 100 mM Sodium pyruvate solution. The total number of cells was counted by trypan blue staining with hematocytometer. The average purity as assessed by flow cytometry was more than 85%.

**RNA extraction and library preparation.** Total RNA was extracted from using RNeasy Mini kits (Qiagen) per manufacturer's instruction, with an additional purification step by on-column DNase treatment using the RNase-free DNase Kit (Qiagen) to ensure elimination of any genomic DNA. Total RNA quality and concentration was analyzed using a NanoDrop 1000 spectrophotometer (ThermoFisher Scientific) and verified using Qubit meter (Invitrogen). All total RNA samples were subjected to level of degradation measurement using Agilent 2100 Bioanalyzer (Agilent Tech Inc.) and RIN values for all samples were ≥9.0. Five hundred nanogram of total RNA from CD8+ or CD4+ T cells was used to generate RNA-seq libraries using TruSeq Stranded mRNA HT kit (Illumina, UK) per manufacturer's instruction. Briefly, RNA was purified and fragmented using poly-T oligo-attached magnetic beads using two rounds of purification followed by the first and second cDNA strand synthesis. Next, cDNA 3′ ends were adenylated and adapters ligated followed by 11 cycles of library amplification. The libraries were size selected using AMPure XP Beads (Beckman Coulter) purified and their quality was checked using Agilent 2100 Bioanalyzer. Samples were randomized to avoid batch effects and multiplexed libraries were run on 12.5 lanes of the HiSeq 2500 platform (Illumina, BRC, Imperial College) to generate 100 bp paired-end reads.

**RNA-seq analysis.** An average depth of 48 M reads per sample was achieved. Sequencing adapters were removed using Trimmomatic (v.0.36) and the reads quality was checked using FastQC (v.0.11.2) before and after trimming. Reads were aligned to the human genome (GRCh38.primary_assembly.genome.fa; annotation: gencode.v25.annotation.gtf) using tophat2 package (v.2.1.0: -b2-sensitive,--library-type fr-firststrand). The average mapping percentage of 92.5% was achieved and the average number of properly paired reads was 42.5 M (approximately 89%). Mapping quality, read distribution, gene body coverage, GC content, and rRNA contamination, were checked using picard (v.2.6.0) software. Gene level read counts were computed using HT-Seq-count (v.0.6.1, annotation: gencode.v25. annotation.gtf)[58] with strict "-m intersection-strict" mode. Genes with fewer than ten aligned reads across all samples were filtered out as lowly expressed genes, keeping 16,874 expressed genes out of 58,037 total genes. A matrix of fragments per kilobase of exons per million reads mapped (FPKM) counts was calculated using Rsubread (v.1.30.5)[59] and edgeR (v.3.22.3)[60] packages. Differential gene expression analysis between groups was performed using DESeq2 (v.1.14.1) and significantly differentially expressed genes were reported using fold-change at 1.5 times and below 1% Benjamini–Hochberg (BH) adjusted P-value. In order to visualize the similarities between samples, hierarchical clustering and PCA were performed using pcaExplorer (v.2.6.0, https://github.com/federicomarini/pcaExplorer) and pheatmap (v 1.0.10, https://CRAN.R-project.org/package=pheatmap) packages respectively. Volcano plots of differentially expressed genes were generated using ggplot2 (v.3.0.0, https://cran.r-project.org/web/packages/ggplot2/index.html) package. All raw RNA-seq data processing steps were performed in Cx1 high-performance cluster computing environment, Imperial College London. Further analyses were conducted in R/Bioconductor environment v.3.4.4 (http://www.R-project.org/).

**Pathway analysis**. Differential expressed genes were analysed with Cytoscape ClueGo v.2.3.3. (http://apps.cytoscape.org/apps/cluego) based of Gene Ontologies (Biological Processes, Molecular Functions, Immune System Process), Interpro, KEGG, Reactome and Wiki Pathways. Terms were called enriched based on maximum P-value of 0.05 and a minimum of 3% overlap. Go Term Fusion and grouping were performed. Enriched groups were further ranked according to the group, Bonferroni's step-down-adjusted P-value. Group lead terms were defined inside each group as the term with the lowest adjusted P-value of enrichment. Visualization of pathways within datasets was done using the Cytoscape (V3.6.0) software. Targeted metabolic pathway enrichment analysis was performed using Cytoscape Reactome Functional Interaction app. Pathways with a maximum false discovery rate of 0.05 were retained as significantly enriched.

**GSVA and WGCN analyses**. To identify pathways correlating with disease activity, GSVA and WGCNA were applied. T cell signatures were correlated with 84 clinical criteria including multiple clinical scoring systems (ACR, BILAG, Physicians Global Assessment, SLEDAI, SLICC), blood parameters (e.g., anti-nuclear, anti-dsDNA, and anti-cardiolipin antibodies, leukopenia), clinical manifestations (lupus nephritis classification, other organ involvement etc), and treatments (azathioprine, hydroxychloroquine, and mycophenolate mofetil etc).

GSVA v1.30.0 was utilised to access whether specific biological pathways or signatures were significantly enriched between active vs. inactive SLE patients. A priori defined gene set was derived from MSig database (http://software.broadinstitute.org/gsea/msigdb). In brief, a matrix of normalized, log-transformed expression data was generated, and all genes were ranked according to the level of expression. Next, enrichment scores (GSVA scores) were calculated non-parametrically using a Kolmogorov Smirnoff (KS)-like random walk statistic for each gene set in a particular sample. Significance difference in enrichment scores between clinical categories was calculated using a chi-squared test and categories with P-values less than 0.05 were considered significantly enriched. GSVA enrichment score for gene set involved in type I IFN signaling (http://software.broadinstitute.org/gsea/msigdb/cards/REACTOME_INTERFERON_ALPHA_BETA_SIGNALING) were used to validate qPCR-based ISM score to show high correlation ($r^2 = 0.9132$ Pearson Correlation Coefficient). To investigate the metabolic pathways that could be influenced by type I IFN signaling, GSVA enrichment score for each gene set related to metabolism from BIOCARTA, KEGG, and REACTOME databases (http://software.broadinstitute.org/gsea/msigdb) was correlated to GSVA enrichment score for the gene set involved in type I IFN signaling using Pearson's r.

WGCNA version 1.68 was used to identify highly correlated genes in each immune cell subset. A matrix of normalized, log-transformed expression data was generated and used as input for WGCNA analysis. A soft thresholding power was chosen on the basis of the criterion of approximate scale-free topology ($P = 15$ for CD4$^+$ T cell dataset and $P = 16$ for CD8$^+$ T cell dataset). Gene networks were constructed, and modules identified from the resulting topological overlap matrix with a dissimilarity correlation threshold of 0.01 to merge module boundaries and a specified minimum module size of $n = 30$. Modules were summarized as a network of modular eigengenes, which were then correlated with a matrix of clinical variables and the resulting correlation matrix visualized as a heat map. Significance of correlation between a given clinical trait and a modular eigengene was assessed using linear regression with Bonferroni adjustment to correct for multiple testing. The biological relevance of gene groups composing modules identified by co-expression analysis was further investigated using STRING v.11 database (https://string-db.org/) to interrogate protein–protein interactions together with enrichment for KEGG pathways and GO terms (Biological process).

**qPCR and ISM score**. RNA samples were converted to cDNA using iScript cDNA Synthesis Kit (Bio-Rad, USA) as per manufacturer's instructions. cDNA was then diluted in EB buffer (Qiagen) to a final concentration of 2.5 ng/μl. Real-time qPCR reactions were conducted using Power SYBR Green Master Mix (Applied Biosystems, USA) and the probe specific primers are listed in Supplementary Table 4. For each PCR reaction, 5 ng cDNA and 0.4 μM primers were used followed by 8 μl of master mix. Each sample was assayed in triplicate. A non-template water control was included for each master mix on all runs. The ViiA 7 System machine (Life Technologies) was used for measuring gene expression. The PCR procedures included a hot start at 95 °C, 10 min and 40 cycles of 95 °C, 25 s; and 60 °C, 60 s. Melt curves were examined for each gene for a single peak. Relative gene expression levels were calculated using $2^{-\Delta\Delta CT}$ method and obtained by normalizing to house-keeping genes *18S* ribosomal RNA *(rRNA)*, *ACTB* (β-actin), or *HPRT1* (Hypoxanthine guanine phosphoribosyl transferase 1). Adapted from Kennedy et al. the ISM score was calculated using expression values from the *CMPK2*, *EPSTI1*, and *HERC5* genes and normalised using the housekeeping gene *HPRT*[27]. The ISM score was calculated from the mean of the *CMPK2*, *EPSTI1*, and *HERC5* cycle threshold (Ct)—*HPRT* Ct (ΔCt) values and multiplied by −1 to give the correct directionality of relative log2-scaled expression. Relative frequency distribution of ISM values in HC and SLE patients was computed in order to define IFN-Neg SLE patient as having ISM < 0.5 and IFN-High SLE patients as ISM > 1.5.

**In vitro type I IFN treatment and T cell stimulation**. PBMCs or CD8$^+$ T cells were isolated from buffy coat cones (NHSBT) from HC as described above. PBMCs

($5.0 \times 10^5$ per well) or CD8$^+$ T cells ($2.0 \times 10^5$ cells per well) were plated into U-bottom 96-well plates in CM containing 10U/ml of recombinant human IL-2 (Peprotech). Cells were treated with human universal Type I IFN (Human Interferon Alpha A/D [BglII]; R&D System, USA; 1000U/ml or other doses as indicated) or left untreated. As indicated cells were either treated with Gibco Dynabeads Human T-Activator CD3/CD28 (beads to cell ratio 1:4) or left untreated. Cells were then incubated at 37 °C in a humidified atmosphere containing 5% CO$_2$ for 2 or 7 days with change of CM at day 3. At the end of the culture, cells were analysed by flow cytometry or cell-sorted and analysed using the extracellular metabolic flux analyser (see below). For the re-stimulation experiments, CD3/CD28 beads were removed from PBMCs culture at day 7 and cells were left to rest overnight. Afterwards, sort-purified CD8$^+$ T cells were re-plated at $2.0 \times 10^5$ cells per well and re-stimulated with Gibco Dynabeads Human T-Activator CD3/CD28 (beads to cell ratio as indicated). After 4 or 24 h of treatment, cells were stained for activation markers and analysed by flow cytometry. IFNγ, TNFα, and Granzyme B cytokines in the supernatant were measured using commercial ELISA kits as listed in Supplementary Table 4 according to manufacturer's protocol.

**Flow cytometry analysis**. The list of antibodies used for flow cytometry is shown in Supplementary Table 4. To exclude dead cells from staining, Live/dead Fixable Aqua stain kit (Molecular Probes, Life Technologies) was used according to the manufacturer's instructions. Staining was performed in the presence of a saturating concentration of Fc Receptor Blocking Solution using Human TruStain FcX™ (Biolegend). For intracellular protein staining, cells were incubated in 1× of BD Fix/Perm™ buffer (BD Biosciences) for 20 min at 4 °C. Cells were washed with 1× of BD Perm/Wash™ buffer (BD Biosciences) and incubated with antibodies for 30 min. After washing with BD Perm/Wash™ buffer, cells were resuspended in FACs buffer (PBS supplemented with 0.1% BSA, 2 mM EDTA, and 0.09% Sodium azide).

Mitochondrial phenotype was assessed using mitochondrial specific fluorescent probes (Supplementary Table 4). $5 \times 10^5$ cells were suspended in 200 μl of CM and treated with mitochondrial probes at indicated concentrations. Cells were incubated for 30 min at 37 °C in a humidified atmosphere containing 5% CO$_2$ before harvesting for flow cytometry analysis. All samples were acquired on a LSRFortessa flow cytometer (BD Biosciences, USA) and data were analysed with FlowJo software, version 10 (Tree Star Inc. Ashland, OR, USA)

**Extracellular metabolic flux analysis**. CD4$^+$ and CD8$^+$ T cells were sorted ex vivo or from day-7 PBMCs cultures using FACs Aria II (BD Biosciences). The metabolic profile of the sorted cells (>95% purity) was analysed using XF Cell Mito Stress kit (Agilent Technologies, USA). Briefly, $3.0 \times 10^5$ cells were suspended in XF medium (Seahorse XF RPMI medium, 2 mM L-Glutamine, 1 mM pyruvate, and 25 mM glucose). Cells were seeded onto a XF96 plate (Agilent Technologies, USA) pre-coated with 22.4 μg/ml of Cell-Tak Cell and Tissue Adhesive (Corning, USA). The culture was equilibrated for 1 h at 37 °C and then OCR (pMole/min) and ECAR (mpH/min) were measured under basal conditions and in response to 1 μM oligomycin, 2 μM fluorocarbonylcyanide phenylhydrazone (FCCP), 1 μM rotenone and 1 μM antimycin A using the XFe96 Extracellular Flux Analyzer. All measurements were repeated at least three times of 3-0-3 min mix-wait-measure cycle. All samples were run in triplicates or quadruplicates. Metabolic parameters were calculated following Gubster et al. publication[61]. For the restimulation experiments of 7-day PBMC cultures, CD3/CD28 beads were removed at day 7 and the cells were left unstimulated overnight. Afterwards, CD8$^+$ T cells were sorted and $2.0 \times 10^5$ cells were suspended in low glucose XF medium (Seahorse XF RPMI medium, 2 mM L-Glutamine, 1 mM pyruvate, and 10 mM glucose), and plated onto an XF96 plate pre-coated with 22.4 μg/ml of Cell-Tak Cell and Tissue Adhesive. After 30 min CD3/CD28 beads (4:1 beads to cell ratio) or phorbol12-myristate13-acetate (PMA)/Ionomycin (final concentration of 100 ng/ml PMA and 500 ng/ml Ionomycin) were directly applied onto the plated cells via the instrument's multi-injection ports. ECAR and OCR were recorded for the duration of the experiment (120 min).

**ATP quantification**. The ATP content of sorted CD4$^+$ and CD8$^+$ T cells was assessed using a bioluminescent ATP assay kit (Abcam). Cells were resuspended at $1 \times 10^6$/ml in phenol free RPMI 1640 and 50 μl cell suspension ($5.0 \times 10^4$ cells) was then added to 50 μl reaction mixture containing ATP monitoring enzyme and nucleotide releasing buffer. Standards were prepared in 10× dilutions (range from 1000 μM to $1 \times 10^{-6}$ μM). Luminescence was measured using a FluoStar Omega plate reader (BMG Labtech, USA). All samples were run in triplicates and ATP level was interpolated from the standard curve.

**Copy number of mitochondria-encoded DNA**. Total DNA was extracted from cell-sorted CD4$^+$ and CD8$^+$ T cells using the DNAzol Reagent following manufacturer's instruction (Invitrogen). The mtDNA copy number was defined as total mtDNA-encoded tRNA (mtDNA-tRNA-F: CACCCAAGAACAGGGTTTGT; mtDNA-tDNA-R: TGGCCATGGGTATGTTGTTA) copies divided by nDNA-encoded B2M (nDNA B2M-F: TGCTGTCTCCATGTTTGATGTATCT; nDNA B2M-R: TCTCTGCTCCCCACCTCTAAGT) copies. For each reaction, 3 μl of sample DNA (1 ng/μl) was amplified in a 10-μl reaction buffer containing 1 μl (5 μM) of each primer and 5 μl of Power SYBR Green PCR Master Mix (Invitrogen). qPCR were run on The ViiA 7 System machine (Life Technologies) as

described above. Simultaneously, 3 µl of PCR grade $H_2O$ were included as negative controls, respectively. Results were expressed as mtDNA copy number as previously reported[62].

**Cell death assay.** PBMCs from SLE patients and HC were kept in culture for 48 h in CM. Cells were washed twice with PBS and then stained for Annexin V and propidium iodide (PI) using the FITC-Annexin V Apoptosis Detection Kit (BD Bioscience, Heidelberg, Germany) according to the manufacturer´s protocol. In brief, PBMCs were resuspended in annexin V binding buffer, FITC-annexin (5 µl) was added to cell suspension containing $2.5 \times 10^5$ cells. The cell suspension was mixed by vortexing gently and then incubated for 15 min at room temperature in the dark. Subsequently, 250 µl of binding buffer and PI (1 µl) were added and cells were analyzed by flow cytometry using BD LSRFortessa (BD Bioscience, Heidelberg, Germany). For restimulation experiments, PBMCs were rested overnight before labelled with 1 µM of CFSE labelling. Cells were then re-plated at $2.0 \times 10^5$ cells per well and re-stimulated with Gibco Dynabeads Human T-Activator CD3/CD28 (1:4, 1:8, or 1:16 beads to cell ratio as indicated). After 3 days, cells were stained for PB-annexin V (1:100 dilution) and analysed by flow cytometry.

**Structured illumination microscopy imaging.** Isolated $CD8^+$ T cells from IFN-Neg ($n = 3$) and IFN-High ($n = 3$) SLE patients were seeded on poly-L-lysine coated high precision cover glasses (1.5H, Marienfeld GmbH & Co.) and subsequently fixed in 4% formaldehyde (w/v) plus 4% sucrose (w/v) in 50% PBS (GIBCO) for 20 min at room temperature. Lymphocytes were then permeabilized with 0.1% Triton X-100 in PBS (5 min). Fish skin gelatine (Sigma, 0.2% w/v) was used throughout to reduce nonspecific antibody binding. Fixed lymphocytes were exposed to primary antibodies against Tom20 (FL-145, sc-11415, Santa Cruz Biotechnology, Inc) and against CD8 alpha (ab199016, Abcam) at 4 °C, overnight. After washing, lymphocytes were labelled with goat anti-rabbit Alexa 488 (Invitrogen) and goat anti-mouse Alex 568 (Invitrogen), as appropriate, for 1 h at room temperature. Nuclei were stained with DAPI. Slides were mounted with antifade medium Vectashield H-1000 (Vector Labs). Lymphocytes were visualized by SR-SIM performed with a Zeiss Elyra S.1 (Carl Zeiss Microimaging) using a Plan-Apochromat 63×/1.4 oil lens and immersion oil (Zeiss Immersol, 518 °F/30 °C). Images were acquired using five phase shifts and three grid rotations, with a z step size of 0.1 µm, and captured with a sCMOS camera (pco.Edge 4.2). The following laser and filter setup was employed: Alexa 568 was excited using a 561 nm laser and emitted light collected with a BP 420–480 + BP 570–640 + LP 740 filter; Alexa 488 was excited using a 488 nm laser and emitted light collected with a BP 420–480 + BP 495–550 + LP 650 filter; DAPI was excited using a 405 nm laser and emitted light collected with a BP 420–480 + BP 495–550 + LP 650 filter. Images were reconstructed using ZEN 2012 SP4 (Black) software, version 13.0.2.518 (Carl Zeiss Microimaging). Channel alignment was achieved by imaging a multi-colored bead slide with the same image acquisition settings. 3-D reconstructions were obtained using Imaris 9.5.1 software (version 9.5.1, Bitplane AG). Imaris "Cells" tool was used to identify $CD8^+$ cells and generate three-dimensional surfaces of the mitochondrion through thresholding of the TOM20 fluorescent channel. Surface renderings less than $0.025 \ \mu m^3$ in size were considered background and excluded. The mitochondrial images were also scored by two blinded researchers (range of 14–43 mitochondrial images per sample; three IFN-Neg and three IFN-High SLE patients). Percentages of $CD8^+$ T cells with fused mitochondria was calculated as total number of cells scored as fused divided by total number of cells imaged in each sample.

**NAD+/NADH assay and in vitro NMN treatment.** $CD8^+$ T cells were isolated from buffy coat cones (NHSBT) from HC using $CD8^+$ T Cell Isolation Kit (Myltenyi Biotec) according to the manufacturer's protocol. $CD8^+$ T cells ($2.0 \times 10^5$ cells per well) were plated into U-bottom 96-well plates in CM containing 10U/ml of recombinant human IL-2 (Peprotech) and Gibco Dynabeads Human T-Activator CD3/CD28 (beads to cell ratio 1:4), with or without Type I IFN (Human Interferon Alpha A/D; R&D System, USA; 1000U/ml). Cells were then incubated at 37 °C in a humidified atmosphere containing 5% $CO_2$ for 7 days. At day 3, cells were supplemented with or without 1 mM NMN (Sigma-Aldrich). At day 7, CD3/CD28 beads were removed from the culture and cells were left to rest overnight. $CD8^+$ T cells were positively selected using Dynabeads FlowComp Human CD8 Kit (Invitrogen) to achieve over 99% purity and then used for: i) mitochondrial respiration response to TCR (beads to cell ratio 4:1) and PMA/Ionomycin (final concentration of 100 ng/ml PMA and 500 ng/ml Ionomycin) stimulation using the extracellular metabolic flux analyser as described above. ii) NAD+/NADH ratio: $2.0 \times 10^6$ $CD8^+$ T cells were lysed and the NAD+/NADH ratio quantified using a specific kit (Abcam, ab65348) following the manufacturer's instructions. All OD450 nm values (microplate reader FluoStar Omega, BMG Labtech, USA) were corrected to the blank controls. Using the OD450 nm values the NAD+/NADH ratio was calculated as: (Total NAD+/NADH) – NADH/NADH. iii) cell viability: $CD8^+$ T cells ($2.0 \times 10^5$ cells per well) were re-stimulated with Gibco Dynabeads Human T-Activator CD3/CD28 (beads to cell ratio 1:2). After 24 h, cells were stained with MitoSOX Red, and after 3 days with Annexin V/PI as described above.

**Statistical analysis.** Data were analysed using GraphPad Prism software, version v.9.0.1 (GraphPad Software, Inc. La Jolla California, USA). Unless indicated, graphs show mean and SEM for each group. One-way ANOVA, and Wilcoxon matched-pairs signed rank test were used to calculate significant differences between multiple or two treatments, respectively. Significance is based upon a $P$ value less than 0.05.

**Reporting summary.** Further information on research design is available in the Nature Research Reporting Summary linked to this article.

## Data availability

RNAseq data have been deposited in the Gene Expression Omnibus repository under accession code GSE97263 and GSE97264. Microscopy image datasets are available from the corresponding author upon request. All the other data are available in the main text or the supplementary materials. Source data are provided with this paper.

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

## Acknowledgements

This work was supported by the Wellcome Trust (Grant reference number: 108008/Z/15/Z) (to M.B.) and by the National Institute for Health Research (NIHR) Biomedical Research Centre based at Imperial College Healthcare NHS Trust and Imperial College London. The views expressed are those of the author(s) and not necessarily those of the NHS, the NIHR or the Department of Health. L.T. is supported by a Development and Promotion of Science and Technology Talents Project (DPST) from Thailand. For the purpose of Open Access, the author has applied a CC BY public copyright licence to any Author Accepted Manuscript version arising from this submission. We thank S. Lewis, C. Shen, A.F. Doyle, L. Stephens, and A. Gilmore for technical support, N. McKenna and R. Santiago for clinical assistance, C. Heiss and G. Bantug for critical discussion and technical help.

## Author contributions

N.B., L.T., V.G., and G.S.L. performed the experiments; A.S. and C.W. performed the imaging studies; L.L. and T.D.C. helped with the collection of patient samples; M.C.P. and J.B. assisted with data analysis and interpretation; G.S.L. and M.B. supervised and conceived the project; N.B. and M.B. wrote the paper. All authors commented on the manuscript.

## Competing interests

The authors declare no competing interests.
