## [Peer Review File · Nature Communications]

REVIEWER COMMENTS

Reviewer #1 (Remarks to the Author):

1. The most critical issue the authors did not address is the mechanistic link between type I interferon and the mitochondrial phenotype. The reduction in mitochondrial function and oxidative phosphorylation can explain the changes in cytotoxic phenotype of CD8+ T cells in figure 5, but how does type 1 interferon causes these mitochondrial changes is not clear.
2. Higher mitotracker deep red in the IFN-high CD8 T cells shown in Figure 2B cannot solely be interpreted as having more active mitochondria. Mitotracker deep red is still a mitochondrial stain, while its accumulation is dependent upon active membrane potential. On the contrary to the above interpretation, these IFN-high CD8 T cells are functionally less competent as the authors have demonstrated with the Seahorse results in Figure 3A. Also, morphologically, the more elongated mitochondria for IFN-high CD8 T cells as shown in Figure 2E can suggest the possibility of failure to fissure or defect in clearance through mitophagy. These could also contribute to functional defects in mitochondria. This interpretation is more consistent with the data presented in figure 3 and 4 – that is that the mitochondria are functional inactive.
3. In vitro T cell culture IFN-alpha results shown in figure 4 are not completely compatible with the results of those found in lupus patients in figure 3. Despite having an increase in mitochondrial mass as shown with higher mitotracker dye, the cells cultured with IFN-alpha are functionally more active with higher basal OCR, which is the opposite of ex vivo results (IFN-high T cells from lupus patients have lower basal OCR). Even though IFN-alpha cultured T cells seem to have lower SRC numerically in figure 4C and 4D, it was due to the fact that they have higher basal OCR and exact the same maximal OCR. Therefore, these data should not be used to interpret as lower bioenergetics impairment of mitochondria.
4. Data in figure 2 are mostly replicative of those published by the Perl lab- true they are extended to T cell subsets.
5. Control samples from people with other IFN signature high diseases should be included.
6. More detailed cytotoxic functional studies are needed to confirm the conclusions.

Reviewer #2 (Remarks to the Author):

Buang et al.

In this paper the authors address transcriptional differences that they have found in circulating CD4 and CD8 T cells from patients with SLE. They show that the cells segregate into two populations, which they refer to as SLE-1 and SLE-2, with the former cells more strongly expressing Type 1 IFN signature genes, plus genes in other pathways. In contrast, mRNAs from mitochondrial encoded genes are expressed less strongly in the SLE-2 cells; as were those for TCA cycle enzymes, and electron transport chain components, suggesting a mitochondrial alteration associate with the Type 1 IFN signature. This correlated with increased mitochondrial size, membrane potential, fusion, and diminished OCR, but only in CD8 T cells, not CD4 T cells. These effects were to some extent superimposable on T cells from healthy individuals by culturing in Type 1 IFN. In ex-vivo cells and in vitro, this resulted in an increase in cell death that the authors suggest may be a major contributor to autoimmunity due to increased autoantigen load.

Specific concerns:

- 1) My understanding of mitochondrial biology is that the image in Fig. 2e indicates that mitochondria are fused in SLE2 patients, compared to fissured in SLE1 patients. There are significant implications for this in terms of expectations of cellular metabolism as measured by Seahorse, and in terms of cellular function. However, these issues are not discussed at all, but need to be.

2) Is this change in mitochondrial morphology observed in the cells stimulated with type 1 IFN in vitro?

3) What percentage of the cells examined had fused mitochondria.

4) I would like to know the extent to which cells in SLE2 vs. SLE1 patients show signs of recent activation – as indicated by flow cytometric evaluation of the CD4 and CD8 T cells for activation markers, or by analysis of the RNAseq for expression of genes linked to recent activation. Are the cells in the SLE2 patients more or less effector like? Are they transcribing IFN γ or Perforin encoding genes?

5) What is the cause of increased cell death in the CD8 T cells from the SLE2 patients? Is apoptotic cell death considered to be as important as necrotic cell death in the pathogenesis of SLE?

6) Is it the authors' assumption that the type I IFN signature is derived from all of the cells or a subset of the cells? (also see #4 above) This kind of information could be obtained using Prime Flow to measure the expression of genes that are part of the signature. If only a percentage of the cells are positive, perhaps these are the cells that are dying ex-vivo?

7) The concluding sentence of the abstract, that "chronic type 1 IFN exposure contributes to lupus pathogenesis by promoting CD8+ T cell dysfunction via metabolic rewiring", is supposition.

Point-to point reply.

We are most grateful for the constructive and positive review of our manuscript, and for the opportunity to submit a revised manuscript for further consideration.

Reviewer #1

The most critical issue the authors did not address is the mechanistic link between type I interferon and the mitochondrial phenotype. The reduction in mitochondrial function and oxidative phosphorylation can explain the changes in cytotoxic phenotype of CD8⁺ T cells in figure 5, but how does type 1 interferon cause these mitochondrial changes is not clear.

As pointed by the Editor and both Reviewers, a clear mechanism linking type 1 interferon to the downstream effects was lacking in the original manuscript. To address this point in the revised version of the paper we present new data demonstrating that **IFN α exposure increases NAD⁺ consumption in activated CD8⁺ T cells causing reduction in cell viability through changes in mitochondrial respiration**. These data are presented in the new figure 6 and in pages 10-11. In brief this is the workflow that led us to this new hypothesis:

- Nicotinate/nicotinamide metabolic pathway was found to show the highest correlation ($R^2=0.6562$; $P<0.0001$) with type I IFN signature in the transcriptomic analysis of our lupus cohort (new Figure 6a).
- The expression of NAD⁺ consuming enzymes (CD38, PARP9, PARP10 and PARP12) is significantly increased in CD8⁺ T cells from IFN-High SLE patients (new Figure 6b).
- TCR-stimulated CD8⁺ T cells from healthy donors increase the surface expression of CD38 (new Figure 6c) and the transcript for PARP9 and PARP10 (new supplementary Fig12a) after IFN α treatment, indicating a downstream effect of IFN α on the nicotinate/nicotinamide metabolic pathway.

Based on these data, we reasoned that the increase in NAD-consuming enzymes was triggered by type I IFN signalling and could lead to NAD⁺ depletion in CD8⁺ T cells. Consistent with this hypothesis, we observed a significant decreased NAD⁺/NADH ratio after IFN α treatment (new Figure 6d).

We have substantiated these observations by showing that:

- NAD⁺ boosting strategy with nicotinamide mononucleotide (NMN) can rectify the downstream effects of IFN α stimulation (i.e. increased NAD⁺/NADH ratio; new Figure 6e).
- NAD⁺ supplementation with NMN increases OCR measured after TCR re-stimulation indicating increased mitochondrial capacity (new Figure 6f).
- The decreased cell viability with IFN α treatment is reversed with NMN treatment (new Figure 6g), indicating a key role of NAD⁺ level to maintain cell viability in IFN α -exposed CD8⁺ T cells.

In summary, the new data indicate that IFN α exposure increases the consumption of NAD⁺ in activated CD8⁺ T cells and reduces the NAD⁺/NADH ratio. This alters the mitochondrial respiration upon re-stimulation and reduces cell viability. The NAD⁺ precursor NMN can rectify these mitochondrial changes and can improve CD8⁺ T cell viability. We have modified the abstract and the discussion and updated the methods/figure legends accordingly.

These new data address the mechanistic link that was missing from the primary submission.

2. Higher mitotracker deep red in the IFN-high CD8 T cells shown in Figure 2B cannot solely be interpreted as having more active mitochondria. Mitotracker deep red is still a mitochondrial stain, while its accumulation is dependent upon active membrane potential. On the contrary to the above interpretation, these IFN-high CD8 T cells are functionally less competent as the authors have demonstrated with the Seahorse results in Figure 3A. Also, morphologically, the more elongated mitochondria for IFN-high CD8 T cells as shown in Figure 2E can suggest the possibility of failure to fissure or defect in clearance through mitophagy. These could also contribute to functional defects in mitochondria. This interpretation is more consistent with the data presented in figure 3 and 4 – that is that the mitochondria are functionally inactive.

The Reviewer is right, and we apologise for the misleading interpretation of the Mitotracker deep red staining. We agree that the increased Mitotracker Deep Red staining may not reflect more active mitochondria. We have revised the manuscript to reflect this (page 7).

3. In vitro T cell culture IFN-alpha results shown in figure 4 are not completely compatible with the results of those found in lupus patients in figure 3. Despite having an increase in mitochondrial mass as shown with higher mitotracker dye, the cells cultured with IFN-alpha are functionally more active with higher basal OCR, which is the opposite of ex vivo results (IFN-high T cells from lupus patients have lower basal OCR). Even though IFN-alpha cultured T cells seem to have lower SRC numerically in figure 4C and 4D, it was due to the fact that they have higher basal OCR and exact the same maximal OCR. Therefore, these data should not be used to interpret as lower bioenergetics impairment of mitochondria.

The Reviewer is correctly pointing out the difference in the basal OCR between the *ex vivo* results from the lupus patients and *in vitro* T cell cultures. However, we would like to point out that the IFN α -treated CD8⁺ T cells, despite the higher basal OCR, are bioenergetically impaired as demonstrated by our findings presented in Figure 4d (i.e. reduced OCR and thus less efficient mitochondrial respiration upon TCR restimulation) and these data are consistent with *ex vivo* results from the IFN-High lupus T cells and the lower SRC. We therefore think that the use of our combined *in vitro* conditions is justified as the *in vitro* cells mimicked closely the *ex vivo* observations from the patient cells. It is worth highlighting that lupus patients tend to be leukopenic and thus some of the mechanistic studies described in the paper that require a relatively high number of cells (e.g. In-Seahorse restimulation analysis) cannot be conducted using patient samples.

4. Data in figure 2 are mostly replicative of those published by the Perl lab- true they are extended to T cell subsets.

We are fully aware of the pioneering work conducted by Prof Perl's lab and we have cited all his work in our paper and discussed our results within the context of his previously published work. However, as noted by the reviewer, our findings do not just replicate the previous observations in more details (e.g. CD8⁺ T cell subsets) but, more importantly, provide, for the first time, a link with the type I IFN signature, something that has never been reported until now. Furthermore, in response to the Reviewer's first point, we have now consolidated this link by providing a mechanism (NAD⁺ consumption) between the type I IFN signature and the mitochondrial changes. The limitations of our study are also discussed in the revised manuscript.

5. Control samples from people with other IFN signature high diseases should be included.

We appreciate the point raised by the Reviewer, but there are no suitable disease controls with a comparable level of type I IFN signature to the one seen in SLE. We include below a brief explanation of why our study design did not include patients with high IFN other than those diagnosed with SLE.

To our knowledge only two other non-infectious conditions have consistently been shown to display a type I IFN signature: Sjögren's syndrome (SS) and Systemic Sclerosis (SSc). Sjögren's syndrome presents several overlaps with lupus and in many patients Sjögren's syndrome is considered secondary to lupus. Thus, patients with Sjögren's syndrome would not represent an independent disease control population. Patients with Systemic Sclerosis also have a type I IFN signature but the current consensus is that this is comparatively much lower than in SLE patients and observed only in a subset of patients, possibly associated with the presence of specific autoantibodies (Assassi et al ARTHRITIS & RHEUMATISM; Vol. 62, No. 2, 2010, pp 589–598). There are other conditions defined as type I interferonopathies (e.g. the Aicardi-Goutières Syndrome) where the type I IFN signature is very high, but these are rare mendelian genetic disorders in children and thus these patients would not be suitable controls. Similarly, patients with juvenile dermatomyositis would not be a valid comparison considering the different age. The evidence of a type I IFN signature in other adult diseases like multiple sclerosis and rheumatoid arthritis remains controversial and weak (Brandon W Higgs, Ann Rheum Dis 2011;70:2029–2036). Consistent with this we found no type I IFN signature in our cohort of rheumatoid arthritis patients. In summary, SLE patients present a unique type I IFN signature that is not recapitulated in other adult autoimmune conditions such as systemic sclerosis, rheumatoid arthritis and multiple sclerosis.

6. More detailed cytotoxic functional studies are needed to confirm the conclusions.

We thank the Reviewer for this suggestion. We agree that the measurement of cytotoxicity is relevant in CD8⁺ T cell biology, but in lupus pathogenesis cell death is a very important readout (see reply to reviewer 2 - point 5). The data presented in our paper show that the metabolic dysfunctions in the IFN α -treated and lupus IFN-High CD8⁺ T cells lead to increased cell death (Figure 5a and 5b) and this would make the interpretation of cytotoxic functional studies difficult and possibly misleading as the number of cells will change during the course of the assay. For this reason, we limited the functional study to the analysis of the percentage CD107a positive cells (Figure 5c and Supplementary Figure 11a). We recognise that this is not adequate to conclude that the cells had reduced cytotoxic ability and for this reason in the paper we only described impaired metabolic fitness and not reduced cytotoxicity.

Reviewer #2

In this paper the authors address transcriptional differences that they have found in circulating CD4 and CD8 T cells from patients with SLE. They show that the cells segregate into two populations, which they refer to as SLE-1 and SLE-2, with the former cells more strongly expressing Type 1 IFN signature genes, plus genes in other pathways. In contrast, mRNAs from mitochondrial encoded genes are expressed less strongly in the SLE-2 cells; as were those for TCA cycle enzymes, and electron transport chain components, suggesting a mitochondrial alteration associate with the Type 1 IFN signature. This correlated with increased mitochondrial size, membrane potential, fusion, and diminished OCR, but only in CD8 T cells, not CD4 T cells. These effects were to some extent superimposable on T cells from healthy individuals by culturing in Type 1 IFN. In ex-vivo cells and in vitro, this resulted

in an increase in cell death that the authors suggest may be a major contributor to autoimmunity due to increased autoantigen load.

Specific concerns:

1. My understanding of mitochondrial biology is that the image in Fig. 2e indicates that mitochondria are fused in SLE2 patients, compared to fissioned in SLE1 patients. There are significant implications for this in terms of expectations of cellular metabolism as measured by Seahorse, and in terms of cellular function. However, these issues are not discussed at all, but need to be.

We thank the Reviewer for the accurate review and constructive comments. We have now amended the discussion (page 15) to address this important point raised by the Reviewer.

2. Is this change in mitochondrial morphology observed in the cells stimulated with type 1 IFN *in vitro*?

As suggested by the Reviewer, we have performed mitochondrial imaging analysis using CD8⁺ T cells stimulated *in vitro* with type I IFN alone and observed no significant changes in mitochondrial morphology of the cells treated with IFN α compared to the untreated cells (see figure a below). Additionally, we found no differences in the total mitochondrial volume (see figure b below). Consistent with the metabolic changes described in the paper, these data suggest that type I IFN alone is not sufficient to generate the morphological changes in mitochondria we observed in the IFN-High SLE patients. Unfortunately, due to the current COVID restrictions the imaging facility is not open anymore and we were unable to perform further independent experiments to consolidate the data and add also additional stimuli (TCR stimulation). We therefore prefer not to include the data presented below in the revised manuscript and we hope the Reviewer appreciates the constraints of the current situation.

Mitochondrial morphology of CD8⁺ T cells treated with IFN α for 7 day

Purified CD8⁺ T cells from healthy donors treated with or without 1000U/ml IFN α for 7 days. Cells were then mounted on poly-L-lysine-coated coverslips, fixed and stained for TOM20 (green) as described in Materials and Methods. Representative SIM images derived from maximal projection analysis (a) and total mitochondria volume (b) of one sample per condition, each with 25 cells analysed. Scale bars, 2 μm . Data presented as mean \pm S.E.M. Each symbol represents a cell. (b) Data analysis using Mann-Whitney test.

3. What percentage of the cells examined had fused mitochondria.

To answer the question raised by the Reviewer we have created a scoring system that classified each cell image as: fused mitochondria (individual mitochondria looks long and tubular in shape) or not fused mitochondria (individual mitochondria looks enlarged and round) or unclear. The images were scored by 2 blinded researchers and the averaged percentage of cells with elongated mitochondria in each patient group was calculated. We found that IFN-High SLE patients have higher percentage of CD8⁺ T cells with elongated mitochondria (21.7 ± 7.4) compared to the IFN-Neg SLE (9 ± 3.4). These new data have been added in the revised version of the paper (page 7 and Materials and Methods page 24). We do, however, recognized that a change in mitochondria volume does not equate to a change in mitochondrial morphology and this is why we show some representative images.

4. I would like to know the extent to which cells in SLE2 vs. SLE1 patients show signs of recent activation – as indicated by flow cytometric evaluation of the CD4 and CD8 T cells for activation markers, or by analysis of the RNAseq for expression of genes linked to recent activation. Are the cells in the SLE2 patients more or less effector like? Are they transcribing IFN γ or Perforin encoding genes?

We would like to clarify that the CD8⁺ T cells from IFN-High lupus patients are not more or less effector-like than the cells from IFN-Neg patients. The transcriptomic analysis that led to the division of the lupus cohort into the SLE2 and SLE1 groups included both active and inactive patients (clinical details in Supplementary Table 1). However, all the flow cytometry, metabolic and functional studies were conducted using inactive lupus patients (clinical details in Supplementary Table 3) and we found no difference in the expression of T cell activation markers between IFN-High and IFN-Neg lupus patients. The clinical difference between these two cohorts of SLE patients may have been overlooked by the Reviewer. Furthermore, in supplementary Figure 5d we showed no significant differences in the percentage of CD8⁺ T subsets between the IFN-High and IFN-Neg inactive lupus patients.

5. What is the cause of increased cell death in the CD8 T cells from the SLE2 patients? Is apoptotic cell death considered to be as important as necrotic cell death in the pathogenesis of SLE?

With regard to the Reviewer's first question, we now provide a mechanism linking the mitochondrial dysfunctions and the CD8⁺ T cell death (for details please see reply to point #1 of Reviewer #1)

The role of apoptotic cell death in lupus pathogenesis is well established. Since the pioneering observations from Casciola-Rosen et al (Casciola-Rosen LA *et al* J Exp Med 179(4):1317-1330, 1994; Rosen, A *et al* J Exp Med 181, 1557, 1995) it has been recognised that autoantigens are displayed on the surface of apoptotic cells, not necrotic cells, and an impaired clearance of these cells, as a result of deficiency in opsonic proteins or their receptors, predisposes to a lupus-like disease in humans and mice (Nagata S *et al* Cell 140(5):619-630, 2010). We were the first laboratory to demonstrate the role of complement C1q, a key lupus susceptibility gene in humans, in the development of autoimmunity (Botto *et al* Nat. Genet. 19, 56–59; 1998) and to put forward the 'waste-

disposal' hypothesis where in the absence of complement apoptotic cells fail to be cleared effectively, promoting further autoantibody production and inflammation (Pickering MC *et al* Adv Immunol 76:227-324; 2000). Therefore, the reduced metabolic fitness observed in the CD8⁺ T cells from IFN-High lupus patients would promote autoimmunity by increasing the autoantigen load as illustrated in Fig 7.

6. Is it the authors' assumption that the type I IFN signature is derived from all of the cells or a subset of the cells? (also see #4 above) This kind of information could be obtained using Prime Flow to measure the expression of genes that are part of the signature. If only a percentage of the cells are positive, perhaps these are the cells that are dying ex-vivo?

The Reviewer has raised a very interesting point and provided a very helpful suggestion. We intended to address this important point. However, by the time we got all the reagents from the company the UK was in the middle of the second COVID wave and the lupus patients, who are immunocompromised, were told to shield again and thus we could not collect anymore samples. As the situation is expected to last for several months, we decided to abort the idea for this study. We hope the Reviewer and the Editor will appreciate the constrains of the current situation especially if the research requires samples from patients who have been told not to come to the Hospital.

We have tried to address the point by looking at the scRNA-seq data available in the literature. Though all the studies have reported the presence of ISG expression in CD8⁺ T cells, from the data it is impossible to identify the CD8⁺ T cell subsets equivalent to the ones defined by flow cytometry. At the moment, we can only say there is no evidence that the type I IFN signature is limited to a specific subset of CD8⁺ T cells, but this needs to be confirmed in the future.

7. The concluding sentence of the abstract, that "chronic type 1 IFN exposure contributes to lupus pathogenesis by promoting CD8⁺ T cell dysfunction via metabolic rewiring", is supposition.

We have revised the conclusion of the abstract.

REVIEWERS' COMMENTS

Reviewer #1 (Remarks to the Author):

The authors have tried to address the comments and indeed they have made significant efforts to link IFN to NAD degrading molecules including CD38. Is CD38 listed among the IFN inducible genes? The molecular pathways which are instigated by CD38 in CD8 cells have been clearly demonstrated. Also, the authors have argued that there is no disease control and despite the extensive diatribe, i cannot agree with them.

Reviewer #2 (Remarks to the Author):

The paper is considerably improved by the revisions. In particular, the data about NAD consumption adds new depth to the findings.

Increased PARP expression associated with increased NAD consumption is an indicator of the DNA damage repair response, which fits generally with what the authors are proposing. Indeed, they say that IFN α induces increased ROS production, and ROS can cause DNA damage. These connections should be discussed in more detail, as they provide context for why PARP expression and NAD depletion may be occurring.

Point-to point reply.

We are very pleased that the reviewers have recognised the value of the additional findings included in the revised version.

Reviewer #1

Reviewer #1 (Remarks to the Author):

The authors have tried to address the comments and indeed they have made significant efforts to link IFN to NAD degrading molecules including CD38. Is CD38 listed among the IFN inducible genes? The molecular pathways which are instigated by CD38 in CD8 cells have been clearly demonstrated.

CD38 is not conventionally considered one of the type I IFN inducible genes (ISGs). Previous studies reported that CD38 promoter contains potential binding sites for IFN regulatory factor-1 (IRF-1)¹ and CD38 expression in cancer cell lines can be induced with both IFN α and IFN β exposures². However, the increase of CD38 expression is not specific to type I IFN exposure as CD38 expression is inducible with activation of other pathways such as TCR or IFN γ signalling³. In our study, we think both TCR stimulation and IFN α exposure contributed to the increase of CD38 expression in CD8⁺ T cells of SLE patients.

Also, the authors have argued that there is no disease control and despite the extensive diatribe, i cannot agree with them.

We appreciate the reviewer's point of view, but we remain of the opinion that there are no suitable disease controls. In some analyses we have used cells from rheumatoid arthritis patients as disease controls but realised immediately that they were of limited value as they did not have the same IFN-signature.

Reviewer #2 (Remarks to the Author):

The paper is considerably improved by the revisions. In particular, the data about NAD consumption adds new depth to the findings.

Increased PARP expression associated with increased NAD consumption is an indicator of the DNA damage repair response, which fits generally with what the authors are proposing. Indeed, they say that IFN α induces increased ROS production, and ROS can cause DNA damage. These connections should be discussed in more detail, as they provide context for why PARP expression and NAD depletion may be occurring.

We thank the reviewer for his/her comment. We have now amended the discussion (Page 15-16) and added the following:

“Activation of PARP proteins is known to be induced by DNA damage⁴. Indeed, deficiency in DNA damage response has been described in SLE and other auto-immune diseases. PBMCs from SLE patients showed decreased PARP activity upon UV-irradiation compared to healthy controls⁵. We observed mROS accumulation after IFN treatment and this could have induced DNA damage and triggered the increased PARP expression. Of note, while ISGs alone were not associated with disease activity, genes belonging to mitochondria-induced apoptosis and DNA damage response pathways significantly correlated with

disease activity (Supplementary Fig. 2a) indicating the potential contribution of these pathways to disease pathogenesis. A recent study reported that upon DNA damage, cells became more dependent on OXPHOS than glycolysis as inhibition of OXPHOS, but not glycolysis, induce ATP deprivation and cell death⁶. Thus, the increased PARP expression associated with the increased NAD⁺ consumption in response to DNA damage could have triggered the metabolic shift in the CD8⁺ T cells from IFN-High SLE patients, a change critical for their survival.”

References

1. Shen, M. *et al.* Interferon regulatory factor-1 binds c-Cbl, enhances mitogen activated protein kinase signaling and promotes retinoic acid-induced differentiation of HL-60 human myelo-monoblastic leukemia cells. *Leuk Lymphoma* **52**, 2372-2379 (2011).
2. Bauvois, B. *et al.* Upregulation of CD38 gene expression in leukemic B cells by interferon types I and II. *Journal of interferon & cytokine research : the official journal of the International Society for Interferon and Cytokine Research* **19**, 1059-1066 (1999).
3. Sandoval-Montes, C. & Santos-Argumedeo, L. CD38 is expressed selectively during the activation of a subset of mature T cells with reduced proliferation but improved potential to produce cytokines. *J Leukoc Biol* **77**, 513-521 (2005).
4. Ke, Y., Zhang, J., Lv, X., Zeng, X. & Ba, X. Novel insights into PARPs in gene expression: regulation of RNA metabolism. *Cell Mol Life Sci* **76**, 3283-3299 (2019).
5. Sibley, J.T., Haug, B.L. & Lee, J.S. Altered metabolism of poly(ADP-ribose) in the peripheral blood lymphocytes of patients with systemic lupus erythematosus. *Arthritis and rheumatism* **32**, 1045-1049 (1989).
6. Murata, M.M. *et al.* NAD⁺ consumption by PARP1 in response to DNA damage triggers metabolic shift critical for damaged cell survival. *Mol Biol Cell* **30**, 2584-2597 (2019).